

# Size-selected black carbon mass distributions and mixing state in polluted and clean environments of northern India

Tomi Raatikainen[1], David Brus[1], Rakesh K. Hooda[1,2], Antti-Pekka Hyvärinen[1], Eija Asmi[1], Ved P. Sharma[2], Antti Arola[3], and Heikki Lihavainen[1]

[1]Finnish Meteorological Institute, Helsinki, Finland
[2]The Energy and Resources Institute, Delhi, India
[3]Finnish Meteorological Institute, Kuopio, Finland

*Correspondence to:* T. Raatikainen (tomi.raatikainen@fmi.fi)

**Abstract.** We have measured black carbon properties by using a size-selected Single Particle Soot Photometer (SP2). The measurements were conducted in northern India at two sites: Gual Pahari is located at the Indo-Gangetic plains (IGP) and Mukteshwar at the Himalayan foothills. Northern India is known as one of the absorbing aerosol hot spots, but detailed information about absorbing aerosol mixing state is still largely missing. Previous black carbon mass concentration measurements

are available for this region and these are consistent with our observations showing that refractory black carbon (rBC) concentrations are about ten times higher in Gual Pahari than those at Mukteshwar. Also the number fraction of absorbing particles is higher in Gual Pahari, but individual absorbing particles including their size distributions are fairly similar. These findings indicate that particles at both sites have similar local and regional emission sources, but aerosols are also transported from the main source regions (IGP) to the less polluted regions (Himalayan foothills). Detailed examination of the absorbing particle

properties revealed that they are most likely fractal aggregates, but the exact structure remains unknown.

## 1   Introduction

Absorbing aerosols are warming the global climate, but uncertainties are still significant partly due to the lack of detailed experimental data on aerosol spatial and temporal distributions and their physical properties (Stocker et al., 2013; Bond et al., 2013). Broadly defined black carbon (BC) is typically the main absorbing aerosol component in submicron aerosols and its

radiative effects depend on absolute concentrations and mixing state, which describes how BC is distributed within the aerosol particles (Bond and Bergstrom, 2006; Petzold et al., 2013; Lack et al., 2014). Although total BC mass concentrations can be routinely measured, the information about the mixing state and size distribution of BC is currently limited. Freshly emitted BC can be almost pure elemental carbon, but rapid atmospheric processing leads to mixed particles containing significant mass fractions of other typical aerosol species such as sulphate and organics. The inclusion of non-absorbing components may

cause an increase to BC absorption by a so-called lensing effect, but this also depends on the structure of the particle (e.g., Adachi et al., 2010; Cappa et al., 2012; Peng et al., 2016). Spherical BC core coated by the non-absorbing material is a major simplification for the particle structure. In addition to the direct radiative effect, aerosol water uptake depends strongly on the volume fraction of soluble aerosol species as pure BC is hydrophobic. Some absorbing aerosol particles can act as a cloud



condensation nuclei (CCN), which means that that BC can have an effect on cloud properties (an indirect climate effect) and that BC can participate in cloud processing. Therefore, knowing the BC mixing state is highly important when assessing the climate effects of BC.

Recent development of single particle instruments capable of detecting BC (e.g., Cross et al., 2010; Lack et al., 2014) has
provided detailed information about the BC mixing state. One widely used instrument for this purpose is the Single Particle Soot Photometer, SP2, (Stephens et al., 2003; Schwarz et al., 2006; Moteki and Kondo, 2007) developed by Droplet Measurement Techniques (Boulder, CO, USA). This instrument uses laser induced incandescence technique to detect so-called refractory black carbon (rBC), which is the fraction of the absorbing carbonaceous material that has boiling point close to 4000 K and therefore, emits visible light when heated to that temperature (Petzold et al., 2013; Lack et al., 2014). The rBC mass can be
detected accurately from most particle types (e.g., Slowik et al., 2007; Cross et al., 2010), while determining the size of the particle containing both rBC and non-refractory material requires significant assumptions about the particle properties (e.g., Taylor et al., 2015). These uncertainties dealing with determining the particle sizes are further reflected in calculations of mixing state parameters such as the rBC volume fraction in absorbing particles and the number fraction of absorbing particles.

Due to the significant local and regional emissions and prevailing meteorological conditions, northern India is one of the
global absorbing aerosol hot spots (Ramanathan et al., 2007). The low frequency of rainfall during the winter and spring months allows the accumulation of aerosol pollutions, which can be observed as a brown cloud (Ramanathan et al., 2001, 2007). Although the absorbing dust aerosol is mainly from natural origin, anthropogenic emissions such as biomass burning and road traffic produce large amounts of black carbon. Aerosol concentrations are decreased significantly when the monsoon rains arrive (typically between mid-June and July in northern India). However, it has been suspected that the increased aerosol
absorption could have an effect on the monsoon (e.g., Menon et al., 2002; Bollasina et al., 2008, 2011; Gautam et al., 2009; Lau et al., 2010; Ganguly et al., 2012; D'Errico et al., 2015; Boos and Storelvmo, 2016), which has a great importance for the whole south Asia region. In spite of the potential importance of the absorbing aerosol, there has been little published information about the BC mixing state in India.

The main purpose of this study is to provide new and detailed information about the rBC mixing state in northern India
focusing on two different environments: polluted Indo-Gangetic plains and relatively clean Himalayan foothills. Comparing observations gives us additional experimental information about processes affecting on the transport and uplift of absorbing aerosol from the plains towards Himalayan foothills. Observations are made with a new measurement system where Differential Mobility Particle Sizer (DMPS) is used to size-select ambient particles before measuring rBC properties with a Single Particle Soot Photometer (SP2). This system provides truly size-resolved information about rBC mixing state parameters including
rBC number fractions and rBC volume in each absorbing particle. Also, comparing the DMPS-selected particle size with that from the SP2 measurements gives additional information about particle morphology.





## 2   Methods

### 2.1   Measurement sites

Mixing state of rBC aerosol was measured in northern India in Mukteshwar, Nainital (29.47° N, 79.65° E, 2180 m above sea level) and Gual Pahari, Gurgaon (28.43° N, 77.15° E, 243 m above sea level) during the spring and pre-monsoon season
2014. Figure 1 shows the station locations. The measurements were started in Mukteshwar (9.2.-31.3.2014) and then the instruments were moved to Gual Pahari (3.4.-14.5.2014). Mukteshwar is a relatively clean site at the foothills of the central Himalayas about 2 km above the Indo-Gangetic plains (IGP) and Gual Pahari station is located at the plains close to Delhi where aerosol concentrations are significantly higher (e.g., Hyvärinen et al., 2009, 2010; Komppula et al., 2009; Panwar et al., 2013; Raatikainen et al., 2014; Hooda et al., 2016).

### 2.1.1   Measurement setup

Refractory black carbon (rBC) concentrations and mixing state parameters were measured by a Single Particle Soot Photometer (SP2; Revision C* with 8 channels), manufactured by the Droplet Measurement Technologies (Boulder, CO, USA), which was connected to a Differential Mobility Particle Sizer (DMPS). Details of the DMPS-SP2 measurement setup, data analysis and a series of consistency tests are given in the Supplementary Material. Briefly, the DMPS is composed of a differential
mobility analyzer (DMA) and a Condensation Particle Counter (CPC). The DMA selects narrow particle mobility size ranges from polydisperse ambient particles and the CPC measures their number concentrations. The actual ambient particle size distribution is then inverted from the sequential observations by the user defined routines (Wiedensohler et al., 2012). The SP2 was connected in parallel to the CPC where it measures the number concentrations and sizes of both non-absorbing particles and those containing refractory absorbing material (Stephens et al., 2003; Schwarz et al., 2006; Moteki and Kondo, 2007). Although
mineral dust and certain metal particles are refractory and absorbing, clear evidence of their contribution (e.g. varying ratios between wide and narrow band incandescence signals) were not observed. Therefore, all refractory material is considered to be refractory black carbon (rBC). In practice, the SP2 measures rBC mass (0.3–380 fg) from individual particles and this can be converted to a rBC volume equivalent diameter (70–740 nm when rBC is represented by a compact spherical core with 1800 $\mathrm{kg\,m^{-3}}$ density). The rBC volume equivalent diameter (briefly rBC core diameter) is a commonly used parameter, but it is well
known that the ambient rBC is not necessarily spherical or compact (e.g., Bond and Bergstrom, 2006; Peng et al., 2016). All number and mass concentrations measured by the SP2 were converted to actual size distributions by using the average DMPS inversion (see the Supplementary Material). The other SP2 parameters such as the fraction of particles containing rBC and the average rBC core diameter, which are now obtained as a function of mobility size and time, do not require this inversion.

Current measurement setup has some similarities with those used by Zhang et al. (2016) and Liu et al. (2013), who coupled
a SP2 with a Volatility Tandem Differential Mobility Analyzer (VTDMA) and Hygroscopicity Tandem Differential Mobility Analyzer (HTDMA). The VTDMA measures particle size distributions after exposing size-selected (200, 250, 300 and 350 nm) particles to 300 °C temperature (Zhang et al., 2016). The same size-selected particles are also measured by the SP2 allowing comparison between rBC core size distributions and those measured by the VTDMA. In the HTDMA-SP2 setup used by Liu



et al. (2013), size-selected particles (163 and 259 nm) are exposed to a high RH (∼90 %) and then measured by the SP2. The main advantages of the current DMPS-SP2 setup is that the size resolution is better (30 logarithmic size bins from about 20 nm to 650 nm) and rBC mass and number size distributions can be determined.

Consistency tests showed that the SP2 over counted particles compared with the parallel CPC measurements (Supplementary Material). Multiplying all SP2 concentrations by a factor of 0.82 made the SP2 and CPC number concentrations levels similar with just noise-like variability. Consistency tests also showed that the DMA-selected mobility sizes are in good agreement with the non-absorbing particle sizes measured by the SP2 although a weak dependency on the SP2 temperature was observed. The last consistency test included a comparison between rBC mass concentration with the optically detected (Aethalometer) equivalent BC (eBC) mass concentration from Mukteshwar. Previous studies have shown that the rBC and eBC values can be different (e.g., Raatikainen et al., 2015), but the current values are within measurement uncertainties. These consistency tests show that the instrument setup and the data analysis methods provide accurate size-resolved rBC size distributions and mixing state parameters.

## 3 Results

Any SP2 can measure rBC core mass distributions (i.e. rBC mass concentration as a function of rBC core volume equivalent diameter) with high time resolution, however, the current size-selected measurements give this information for each DMA-selected mobility particle size. Knowing the particle (mobility) size simplifies the calculations especially for absorbing particles, which evaporate when travelling through the laser beam. For those particles, Leading Edge Only (LEO) methods (e.g., Gao et al., 2007; Metcalf et al., 2012; Laborde et al., 2012) can be used to calculate the optical size from the scattered laser light, but the calculations require additional particle position information and the results depend on the assumed particle structure and optical parameters. In the following calculations particle size is assumed to be the DMA-selected mobility diameter, which is the case for spherical particles. However, optical and mobility sizes are compared in Sect. 3.6 to obtain additional information about particle morphology.

Thanks to the significantly improved size resolution, it is possible to examine how much variability is there within each DMA-selected particle size. It is often assumed that each particle size has a narrow unimodal distribution of rBC core sizes, but we show that this is not always the case. Since this level of detail is rarely needed, we will focus on the particle properties averaged for each DMA-selected size. This means that we calculated rBC number and mass size distributions and size-dependent rBC mixing state parameters (number fraction of particles containing rBC and the average volume fractions of rBC and non-refractory material in absorbing particles). These size distribution and mixing state parameters have also variations in time.

### 3.1 Total rBC mass concentration time series

As an example of the measured parameters and their time variations, the total rBC mass concentration time series from the both measurement stations are shown in Fig. 2. Time series of the other parameters, which will be described below, are shown in the



Supplementary Material. Figure 2 shows that the rBC mass concentrations are highly variable and the variability is dominated by their diurnal cycles. Detailed examination of the diurnal variations of the total rBC mass and the other measured rBC mixing state parameters will be given in Sect. 3.5. Figure 2 also shows the clear difference in rBC mass concentration between the site at the polluted Indo-Gangetic plains (Gual Pahri) and the relatively clean site at the Himalayan foothills (Mukteshwar). These

differences will be examined in the following sections.

## 3.2  Size-selected rBC homogeneity

Size-selected measurements of rBC core size distributions can show how homogeneous absorbing particles are. When a sufficiently large mobility size ($\geq$ 200 nm) is selected, multiple rBC core diameter modes appear especially in Mukteshwar. Figure 3 shows the campaign average rBC core number size distributions from particles with 360 nm DMA-selected mobility diame-

ter. This mobility size seem to be optimal for most calculations as it is representative of the accumulation mode and has enough particles for good counting statistics; although the standard deviations for the concentration-dependent parameters such as the average rBC core number size distributions (standard deviation not shown in Fig. 3) are comparable with the corresponding average values, these are mainly caused by the clear diurnal concentration variations seen in Fig. 2. The 360 nm mobility size is also large enough to avoid missing the rBC cores from "thickly coated" particles (about 70 nm detection limit for rBC volume

equivalent diameter). This is the reason why current calculations are limited to mobility sizes larger than ~200 nm.

When Gual Pahari rBC core size distribution is mostly unimodal, that at Mukteshwar is clearly bimodal where the smaller mode is located at about 110 nm and the other dominating mode is at about 210 nm. Changing the DMA-selected diameter to a larger or smaller value does not reveal any additional modes. It should be noted that here the particles larger than about 300 nm are most likely from multiply charged particles (no clear evidence of pure compact rBC; see Sect. 3.6), but their contributions

for the rBC core mean diameter and the total particle number and mass are negligible.

Mukteshwar rBC core number size distribution seems to be bimodal most of the time. The number fraction of the larger rBC particles (those larger than 140 nm from the 85–300 nm core size range) varies between 0.5 and 0.8. The fluctuations are irregular covering several days, and mainly for this reason, there are no significant diurnal variations (not shown). The lack of diurnal variations indicates that the mixing of local and long-range transported (from the Indo-Gangetic plains or aloft) air

masses is not the main reason for the bimodal rBC size distribution (this explains the diurnally varying rBC mass concentration described in Sect. 3.5). Another explanation for the observed bimodal rBC distribution is that the smaller and larger particles are originating from different local sources.

## 3.3  Average rBC size distributions

Since rBC homogeneity within a DMA-selected mobility size is too detailed information for most practical applications, the

following calculations are based on rBC properties averaged over each DMA-selected mobility size. Because rBC core size distributions are not fully unimodal, the average rBC core diameter is calculated based on both particle number (as in Fig. 3) and mass (or volume, which is just the rBC mass divided by the constant density of $1800 \, \mathrm{kg \, m^{-3}}$). The former is simply the number average rBC core diameter and the latter is representative of the rBC core mean mass so that the correct total rBC





mass can be obtained by multiplying the core mean mass by the total number of particles containing rBC. The number fraction of particles containing rBC, $N_{rBC}/N_{total}$, is also calculated for each DMA-selected mobility size by using the total number of observed particles and that for particles containing rBC. Finally, logarithmic rBC number and mass size distributions are calculated (see the Supplementary Material) and described with the total number and mass concentration and geometric (mass) mean diameter and standard deviation.

Figure 4 shows the average rBC core mass and number size distributions from the both measurement sites during the campaign period. The dip at about 300 nm is caused by a gap between high and low gain rBC mass calibration parameterizations. Due to the significant diurnal variations, which will be discussed later, the average mass and number size distributions have standard deviations (not shown) that are proportional to the observed size bin mean values. It is evident from Fig. 4 that the number size distributions are not fully resolved due to the about 70 nm rBC core size detection limit. Therefore, we will focus on the rBC mass size distributions. These have similar shapes except that the concentrations at Gual Pahari are about ten times higher than those at Mukteshwar. Both mass distributions peak at around 210 nm, but these have relatively high concentrations of larger particles especially at Gual Pahari. Large particles are also observed in the number and mass size distributions measured by the DMPS (shown in the Supplementary Material). The large absorbing refractory particles are most likely rBC. Dust particles and rBC have different boiling points, which would mean different wide band to narrow band incandescence ratios (Moteki and Kondo, 2007), but such changes were not seen.

Figure 4 shows the campaign average distributions, but we have also calculated the corresponding time series (rBC mass concentrations time series is shown in Fig. 2 and the other times series are shown in the Supplementary Material). These log-normal mass distributions can be described by the total rBC mass concentrations and the geometric mass mean diameter and standard deviation. Average size distribution parameters and their standard deviations from both measurement sites are shown in Table 1. The diurnal variations of the key rBC size distribution and mixing state parameters are also shown in Sect. 3.5.

The observed total rBC mass concentrations are similar to the typical optically measured equivalent BC (eBC) concentrations. For example, previous long term measurements have shown that the average April and May eBC concentrations in Gual Pahari are 8.5 and 5.7 $\mu g\,m^{-3}$ (Hooda et al., 2016), respectively, and those in Mukteshwar are 1.4 and 1.2 $\mu g\,m^{-3}$ (Hyvärinen et al., 2009). Since our simultaneous SP2 and Aethalometer measurements from Mukteshwar show good agreement between rBC and eBC concentrations (shown in the Supplementary Material), current measurements seem to represent typical pre-monsoon season conditions.

Previous SP2 studies have reported rBC core size distribution parameters from various environments (e.g. summary in Huang et al., 2012), but to our knowledge there are no previously published results from India. However, equally high rBC concentrations are observed in China and there SP2 studies have shown that rBC core peak diameters are close to 220 nm (Huang et al., 2011, 2012), which are in good agreement with the current observations (peaks at about 210 nm). Due to the presence of large (>400 nm) rBC cores, geometric mean diameter is somewhat higher in Gual Pahari than in Mukteshwar or in these previous observations elsewhere. In general, mass mean diameters are relatively similar at least compared with concentrations which vary by several orders of magnitude depending heavily on local emission sources.



### 3.4 Average rBC mixing state

Mixing state can be described by two size-dependent parameters that are directly measured by the DMPS-SP2 system: number fraction of particles containing rBC ($N_{rBC}/N_{total}$) and rBC volume fraction in these particles. For simplicity, it is expected that particles are spherical so that their volumes can be calculated using the mobility diameter (some indirect evidence about the actual particle morphology is given in Sect. 3.6). With this assumption rBC volume fraction is proportional to the rBC core to mobility diameter ratio. Because rBC core diameters show some variability (Fig. 3), we have calculated both number and volume mean rBC diameters ($D_{rBC,N}$ and $D_{rBC,V}$) and their ratios with mobility diameters ($D_m$). The campaign average mixing state parameters and their standard deviations for the 360 nm mobility size are shown in Table 1. Their diurnal variations are examined in Sect. 3.5.

The data in Table 1 shows that a relatively large fraction of particles contain detectable amounts of rBC, but these absorbing particles have low rBC volume fractions in both Gual Pahari and Mukteshwar. As expected, the rBC number fraction is somewhat larger in Gual Pahari (polluted region) than in Mukteshwar (regional background), but the absorbing particles seem to have similar rBC volume fractions. It could have been expected that rBC volume fraction decreases when semi-volatile species such as organics and sulfate condense to existing particles during their transport to Mukteshwar, but this effect is not clearly seen although it may contribute to the observed bimodal rBC core size distribution seen in Fig. 3. It seems that the observed rBC properties are common for the whole region due to the similar emission sources and relatively short times for aging (more likely hours than days). For example, air masses in the upper troposphere or in remote regions can have spent several days without any contact to rBC sources.

There are some previous studies that describe the rBC mixing state with this level of details. It is evident that most particles do not contain detectable amounts of rBC anywhere (e.g., Kondo et al., 2011; Reddington et al., 2013; Dahlkötter et al., 2014; Raatikainen et al., 2015), but the current fractions of rBC-containing particles (46 % and 31 % for Gual Pahari and Mukteshwar, respectively) seem to be the highest so far. For example, our previous results from the Finnish Arctic show that 24 % of the particles contain rBC and this is already a relatively large fraction (Raatikainen et al., 2015). In India, the high number fraction of rBC-containing particles is probably resulting in from the significant local and regional biomass burning emissions. The observed rBC core to particle diameter ratios ($D_{rBC,V}/D_m$ or $D_{rBC,N}/D_m$), which represent rBC volume fractions, are quite low for a fresh aerosol anywhere (e.g., Kondo et al., 2011; Schwarz et al., 2008; Sahu et al., 2012; Metcalf et al., 2012). Our qualitative analysis of particle morphology (see Sect. 3.6) indicates that the absorbing particles are fractal aggregates, which means that the mobility size is significantly larger than the actual volume equivalent diameter, but for the time being mobility size is used as a representative of the particle size.

Mixing state parameters are somewhat size-dependent and this can be parameterized using the size-selected measurements. Fig. 5 shows the averaged size dependent mixing state parameters (number and volume based rBC core diameters and number fractions of particles containing rBC) and simple parameterizations. The lowest particle sizes where the SP2 detection limit has a significant effect on the results have been excluded from the fits (indicated by the smaller marker below ∼200 nm particle size). Also, the largest particle size in Mukteshwar has also been excluded due to the low number of observed particles. In





general, the trends in the rBC mixing state parameters are similar for Gual Pahari and Mukteshwar, which indicates fairly similar local and regional rBC sources.

## 3.5 Diurnal cycles

Figure 6 shows the diurnal cycles of the rBC mass distribution (total mass and geometric mass mean diameter and standard
deviation) and mixing state (rBC core to particle diameter ratio and the number fraction of particles containing rBC for the 360 nm mobility size bin) parameters. The number based diameter ratios and size distribution parameters are not shown, because these have similar diurnal variations with the mass and volume based parameters. The total rBC mass concentrations have significant diurnal variations while those for the mean diameter and distribution width are modest. From the rBC mixing state parameters, which are not directly related to the mass distribution, only the rBC particle number fractions have clear diurnal
cycles while the rBC core to particle diameter ratio is practically constant.

Strong diurnal variability of the optically measured equivalent black carbon and the total particulate mass in general have been observed in our previous studies and by others (e.g., Komppula et al., 2009; Hyvärinen et al., 2009, 2010; Panwar et al., 2013; Raatikainen et al., 2014), and similar diurnal cycles are also seen in the rBC mass concentrations. Aerosol pollutions from local and regional sources accumulate to the shallow boundary layer during late evening and night at the Indo-Gangetic
plains (represented by Gual Pahari). At this time rBC number fractions are also the highest. Aerosol concentrations and rBC number fractions decrease in the morning due to the increased vertical mixing. During the pre-monsoon season, when boundary layer heights typically exceed 2000 m, polluted air masses from Indo-Gangetic plains reach the altitude of Mukteshwar station during afternoons (Raatikainen et al., 2014). This is seen as a significant increase in total aerosol and rBC concentrations, but the other rBC parameters are practically unchanged. This means that transported aerosols have already become similar to
the background aerosols, i.e. lost a fraction of the largest absorbing particles and grown by condensation of a non-refractory material. In general, fresh (Gual Pahari) and background (Mukteshwar) rBC aerosols are relatively similar except that the fresh aerosol has an order of magnitude higher concentration and the number fraction of particles containing rBC is about 50 % larger compared with those from the background site.

## 3.6 Morphology of absorbing particles

Black carbon or soot particles are initially aggregates composed of several primary BC particles, which diameters are in the order of a few tens of nanometers (e.g., Sorensen, 2001). These fresh aggregates can contain some amounts of a non-refractory material, but the fraction increases with time when atmospheric vapours condense to the soot particles and when the particle grow by coagulation. Increasing non-refractory fraction makes these particles more spherical. In addition, aggregates can be compacted when particles absorb water vapour and become droplets (e.g., Zhang et al., 2008; Pagels et al., 2009). As a result,
core-shell structure can be a valid approximation for the aged aerosol, but it is not clear if this is the case in India, where the aerosol is relatively fresh. The SP2 can provide some information about the morphology of the absorbing particles.





First, the SP2 can detect if a particle disintegrates in the laser beam into rBC and non-absorbing fragments. This can happen if the rBC core is close to the particle surface or when rBC is attached to the surface of another particle (Sedlacek et al., 2012; Moteki et al., 2014). However, current experiments showed that such disintegrating particles have negligible concentrations.

There are also studies reporting bare rBC particles (e.g., Huang et al., 2012), but the current rBC core volume equivalent diameters are always well below mobility diameters. However, it is possible that the rBC particles have low densities or that the particles are irregular aggregates. At least qualitative information about the particle shape can be obtained by comparing the DMA-selected mobility and SP2-derived optical sizes. Optical size is based on the measured or reconstructed (see below) intensity of the scattered laser light and a theoretical correction to the scattering accounting for the difference between calibration (ammonium sulfate) and ambient aerosol structures and optical properties. Accurate sizing requires information about particle structure, but clear differences between optical and mobility sizes (at least $\pm$ 10 %) indicate non-spherical particles such as aggregates.

Optical sizes are estimated using Leading Edge Only (LEO) methods (e.g., Gao et al., 2007; Metcalf et al., 2012; Laborde et al., 2012). In earlier studies, the authors use the leading edge of the scattering signal, which is yet unaffected by the evaporation of the non-refractory material, to reconstruct the unperturbed scattering signal. Current method is the same as the method used in our previous study (Raatikainen et al., 2015): here the leading edge is the part of the signal where laser beam intensity is 0.07–3 % of the maximum, and the Gaussian scattering (laser beam) profile and peak position are calculated by averaging those from 100 previous non-absorbing particles. Signal peak height is solved by fitting the Gaussian profile to the signal from the leading edge (e.g., Gao et al., 2007). Although the current LEO analysis suffered from a noise signal (see the Supplementary Material), which reduces the reliability of the results, the difference between the DMA-selected mobility and optical size is clear. Taking the 360 nm mobility size bin as an example, the average LEO-calculated optical size for absorbing particles is only about 220 nm in both Mukteshwar and Gual Pahari (non-absorbing particles have almost equal LEO-calculated optical and mobility sizes). This 220 nm average optical size is quite close to the mean rBC core volume equivalent diameter (Table 1: approx. 180 nm for the 360 nm DMA size), which is indicative of scattering from highly fractal soot aggregates. Such aggregates typically have low volume fractions of non-refractory material, but the exact fractions cannot be determined without detailed information about particle morphology and optical properties.

Zhang et al. (2016) used similar measurement setup (VTDMA-SP2) in northern China about 60 km from Beijing. They have found closely matching LEO-calculated optical and mobility sizes, which indicates spherical particle shape and internally mixed particles with low rBC volume fraction (161 nm mass equivalent rBC core size for 350 nm mobility size). Although the rBC concentrations have similar magnitudes in northern China and India, absorbing particles seem to have different properties most likely due to different sources.

Additional information about aggregate properties can be found by calculating parameters such as effective densities, dynamic shape factors and mass fractal dimensions, which are based on the measured particle mass and mobility diameter (e.g., Zhang et al., 2008; Pagels et al., 2009; Peng et al., 2016). The SP2 can detect the total rBC mass from each particle (e.g., Slowik et al., 2007), but non-refractory mass cannot be determined from irregular particles. Therefore, we will focus on the mass fractal dimension parameter, which can be less sensitive on the possibly missing non-refractory mass. Empirical equa-





tions of the same kind have been often used in cases where particle mass ($m$) and mobility size ($D_m$) are known while the structural details are missing (e.g., Weber et al., 1996; Park et al., 2003; Zhang et al., 2008; Pagels et al., 2009):

$$m = A D_m^{Dfm} \tag{1}$$

Here $Dfm$ is the mass fractal dimension and it is a fitting parameter in addition to the constant $A$. Constant $A$ depends directly

on the absolute particle mass while $Dfm$ is independent of that when the possibly missing non-refractory mass is a constant fraction of the observed mass. $Dfm$ is indicative of the compactness of the particle: values close to 3 mean a sphere and lower values indicate less ideal particle shape. Figure 7 shows the average rBC mass (calculated from the mass mean core volume equivalent diameter) as a function of mobility diameter and a fit (initially log-log scale: $\log(m) = \log(A) + Dfm \log(D_m)$) to the data for both Mukteshwar and Gual Pahari. Both Gual Pahari and Mukteshwar mass fractal dimensions are low (1.72

and 1.85, respectively) indicating highly fractal particle structure (e.g., Pagels et al., 2009). This is in a qualitative agreement with the findings of Coz and Leck (2011) who observed the dominance of freshly formed soot particles with an open branched structure in Sinhagad near Pune in western India during April 2006.

   Mass fractal dimensions were also found for individual measurements using the same fitting method (not shown). The $Dfm$ time series show larger and more irregular variations in Mukteshwar (the 10th and 90th percentiles are 1.38 and 2.22,

respectively) than in Gual Pahari (the 10th and 90th percentiles are 1.48 and 1.95, respectively). As a result, there are no clear diurnal cycles in Mukteshwar where hourly averages ($\pm$ standard deviation) range from $1.76 \pm 0.29$ at 17:00-18:00 LT to $2.00 \pm 0.35$ at 05:00-06:00 LT. The hourly averages in Gual Pahari range from $1.61 \pm 0.15$ at 04:00-05:00 LT to $1.90 \pm 0.20$ at 15:00-16:00 LT, which means small but observable diurnal cycle. Nevertheless, the highly fractal particles with $Dfm$ less than two seem to be dominating at least in Gual Pahari. This indicates slower aging than in Mexico City, where photochemical

activity results in rapid conversion of fresh soot to spherical coated soot (Moffet and Prather, 2009).

## 4   Conclusions

Refractory black carbon (rBC) mass distributions and mixing state parameters were measured using a size-selected Single Particle Soot Photometer (SP2) in northern India during spring 2014. The size-selected results were obtained by connecting a SP2 to the outlet of a Differential Mobility Analyzer (DMA), which classifies particles according to their mobility size.

The measurements were made in a relatively clean regional background site at the Himalayan foothills (Mukteshwar) and at a relatively polluted site close to Delhi (Gual Pahari). To our knowledge, this is the first publication showing size-selected rBC mass distributions and mixing state parameters for this region.

   The measurements show that about 30–50 % of the accumulation mode particles contain observable amounts of rBC. Just as the absolute rBC concentrations, the rBC particle number fraction is higher at the source regions (represented by Gual Pahari)

and lower at elevated altitudes (Mukteshwar). Although literature data about rBC mixing state is limited (e.g., Raatikainen et al., 2015; Dahlkötter et al., 2014; Reddington et al., 2013), the observed number fractions of particles containing rBC are the highest reported so far.





The observed rBC particles are likely to contain a non-refractory material such as sulfate and organics, but the exact volume fractions could not be quantified. Current rBC volume equivalent to mobility diameter ratios (number mean $D_{rBC,N}/D_m$ ~0.5 for both measurements sites) would mean that such spherical particles have lower rBC volume fractions than expected for fresh particles (e.g., Schwarz et al., 2008; Sahu et al., 2012). The rBC diameter to optical size ratio (~0.8 for both measurement sites) is closer to value expected for fresh aerosol, but these calculations are limited by the fact that the optical size is based on assumed optical parameters and spherical core-shell structure. Furthermore, the clear difference between optical and mobility sizes as well as low values of fitted mass fractal dimensions (see e.g., Pagels et al., 2009) indicate that the rBC particles are most likely highly irregular fractal aggregates. In that case the exact calculations of particle size or total material volume are not possible without additional details about particle structure.

Although individual particles seem to be quite similar in Gual Pahari and Mukteshwar, the total rBC concentrations are about ten times higher at the more polluted site, Gual Pahari, than those at the regional background site, Mukteshwar. Also, a larger fraction of the particles contain rBC in Gual Pahari than in Mukteshwar. One explanation for the similarity is that some aerosol sources are common for the whole region (e.g. road traffic, biomass burning and cooking). The other is that a significant fraction of the rBC seen in Mukteshwar can be originating from the densely populated Indo-Gangetic plains represented by Gual Pahari (Raatikainen et al., 2014).

Detailed information about the black carbon mixing state is needed for assessing and improving the performance of climate models in simulating their evolution and radiative effects. SP2 is one of the few instruments that can provide detailed information about the absorbing aerosol mixing state. The accuracy of the mixing state parameters can be further improved by size-selecting the particles before measurements with the SP2; this method is especially suitable for polluted areas where good counting statistics is guaranteed.

*Acknowledgements.* Authors would like to acknowledge the Academy of Finland (project numbers 264242, 268004 and 284536), Academy of Finland Centre of Excellence Program (project number 272041) and KONE foundation for the financial support. We thank to local TERI staff for 24/7 work at Mukteshwar aerosol research station.





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





**Table 1.** Campaign average values describing rBC mass size distributions (total mass and geometric mass mean diameter and standard deviation) and mixing state (number and volume mean core diameters, those normalized by the mobility size ($D_m$), and number fraction of particles containing rBC). The mixing state parameters are calculated for the 360 nm mobility size bins.

| Parameter | Gual Pahari | Mukteshwar |
|---|---|---|
| Total rBC ($\mu g\,m^{-3}$) | $11 \pm 11$ | $1.0 \pm 0.6$ |
| GMD dm/dlog$D_{rBC}$ (nm) | $249 \pm 30$ | $217 \pm 13$ |
| GSD dm/dlog$D_{rBC}$ | $0.246 \pm 0.014$ | $0.221 \pm 0.014$ |
| $D_{rBC,N}$ (nm) | $185 \pm 8$ | $178 \pm 12$ |
| $D_{rBC,N}/D_m$ | $0.51 \pm 0.02$ | $0.50 \pm 0.03$ |
| $D_{rBC,V}$ (nm) | $221 \pm 14$ | $205 \pm 16$ |
| $D_{rBC,V}/D_m$ | $0.61 \pm 0.04$ | $0.57 \pm 0.04$ |
| $N_{rBC}/N_{total}$ | $0.46 \pm 0.12$ | $0.31 \pm 0.05$ |





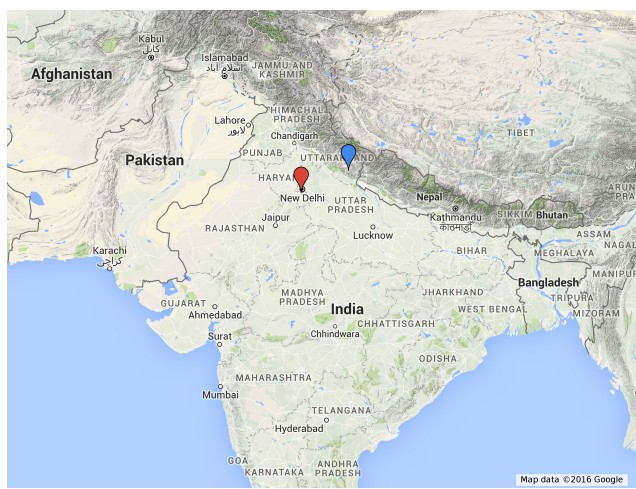

**Figure 1.** Locations of Gual Pahari (red marker) and Mukteshwar (blue marker) measurement stations.

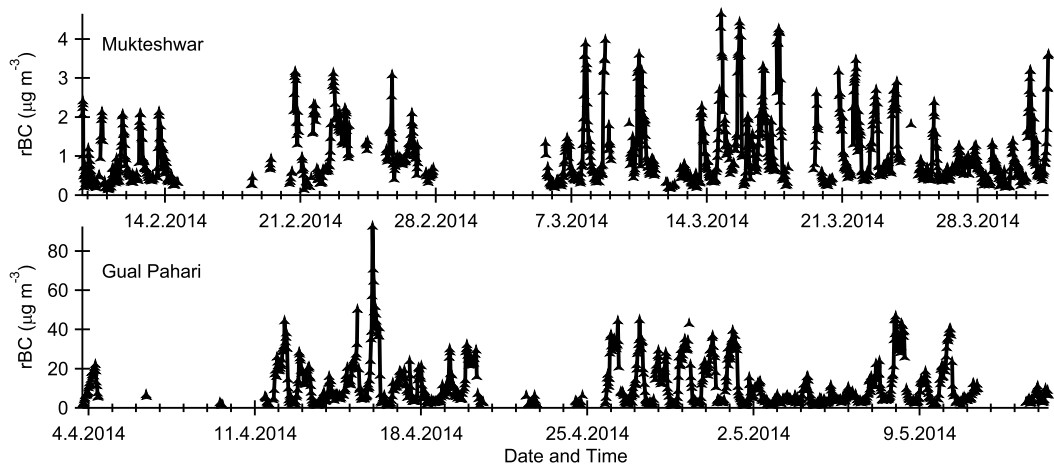

**Figure 2.** Total rBC mass concentration time series from Mukteshwar (top graph) and Gual Pahari (bottom graph).




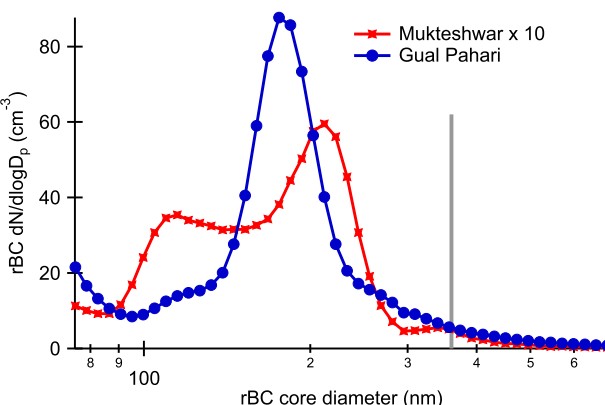

**Figure 3.** Campaign average rBC core number size distributions for the 360 nm DMA-selected mobility diameter (indicated by the vertical gray line) from Mukteshwar (multiplied by a factor of ten) and Gual Pahari. Standard deviations are approximately equal with the average concentration values when concentrations are larger than $1 \ \mathrm{cm}^{-3}$ while smaller concentrations mean increasing standard deviations.

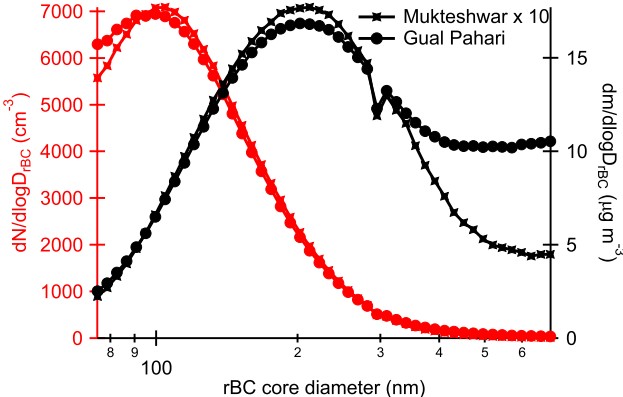

**Figure 4.** Campaign average rBC mass (black color, right axis) and number (red color, left axis) size distributions for Mukteshwar (lines with circles) and Gual Pahari (lines with crosses). Mukteshwar number and mass size distributions have been multiplied by a factor of ten. Standard deviations are approximately 80% (Mukteshwar) or 100% (Gual Pahari) of the bin mean number and mass.





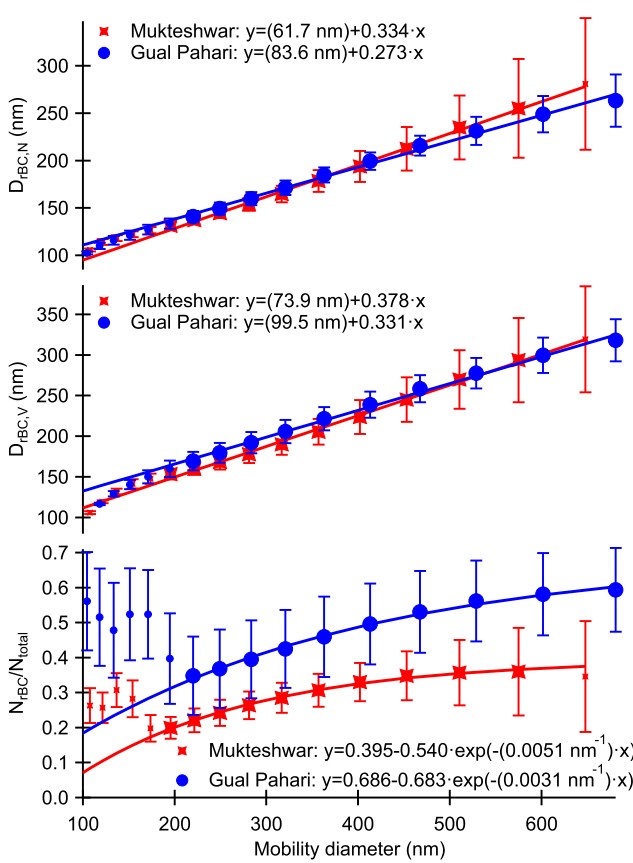

**Figure 5.** Size dependent rBC mixing state parameters for Gual Pahari and Mukteshwar. The upper panel shows the number mean rBC core volume equivalent diameters and the mid panel shows those based on the rBC volume. The lower panel shows rBC particle number fractions. Solid lines are the fits to the data (ignoring points with smaller marker size). Error bars indicate ±1 standard deviation limits.



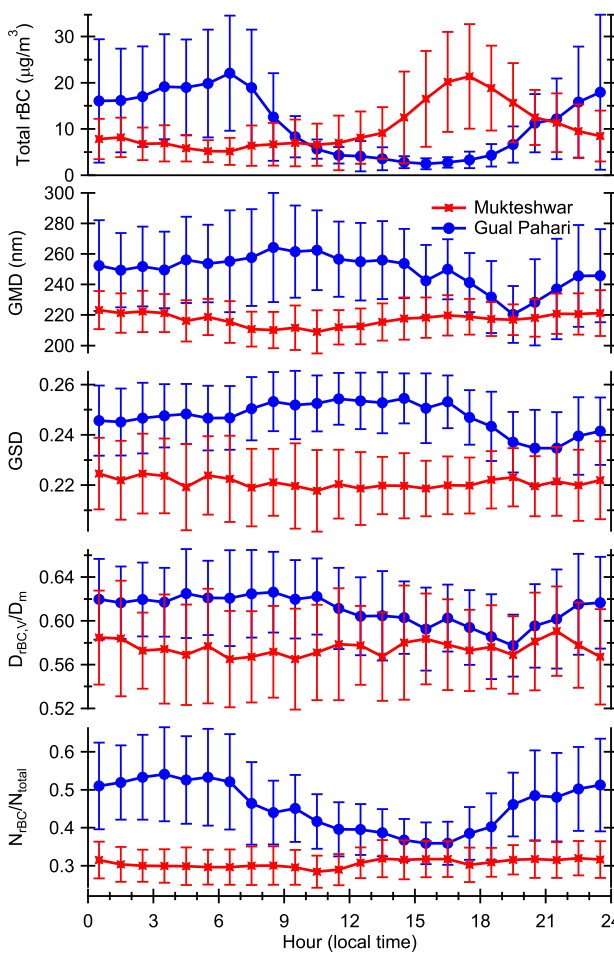

**Figure 6.** Diurnal cycles of rBC core mass size distribution parameters (total mass concentrations and geometric mass mean diameter and standard deviation), rBC core to mobility diameter ratios and fractions of particles containing rBC. The diameter ratios and fractions are calculated for the 360 nm mobility size. Error bars indicate ±1 standard deviation limits. Mukteshwar total rBC mass concentration and its standard deviation have been multiplied by a factor of ten.





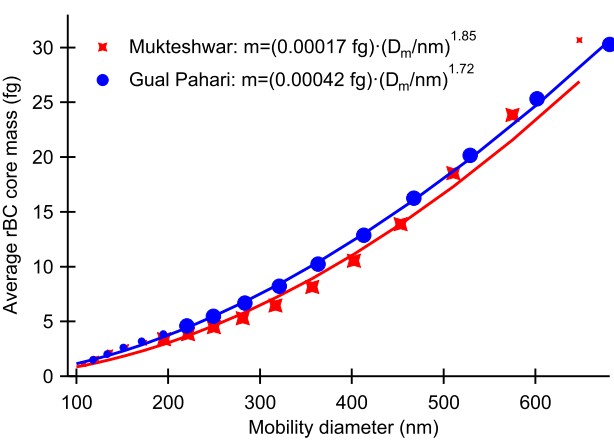

**Figure 7.** Campaign average rBC core mass as a function of mobility diameter and fits to the observations (ignoring points with smaller marker size).