# Peer review of "Size-selected black carbon mass distributions and mixing state in polluted and clean environments of northern India"

_Atmospheric Chemistry and Physics, 2016_

## Referee Comment (RC1) · Anonymous Referee #1 · 17 Jul 2016

I think this paper needs to be more conclusive (too many hypothesis at the moment) before it can be accepted. The writing itself I feel a bit lengthy. The major points in particular:

-the BC sources are not clear, some back trajectory analysis will be helpful. This unclearness goes through all of the texts when discussing if the source is local or transported etc. The discussions on diurnal variation are also weak because of lack of source analysis. With clear source analysis, these discussions should be tidier.

-the main limitation is most of the information is derived from DMA360nm, which could only represent a fraction of total BC. The most populated total particle size may not represent the most populated rBC size, therefore combing the rBC size at other DMA

sizes will be also useful. I would suggest to analyse and compare the rBC information at all DMA sizes together.

-the chamber temperature introduces some instrument bias, I guess the SP2 laser power was affected by this? how have you corrected this regarding the reduced detection efficiency when reduced laser power?

-the BC mixing state as derived from DMPS-SP2 is not clear, did you calculate as rBC size divided by mobility size? I don't think page 7 has explained what you have done sufficiently. This is really important but this only appears in supplement. The uncertainty of this method is largely from the particle morphology, however it is hard to tell without particle mass measurement (though you measured rBC mass but this is not the total BC-containing particle mass). I think the main texts need to address this uncertainty aided by more thorough analysis at different DMA sizes.

-The bimodal mode of rBC under cleaner environment looks interesting which needs more detailed analysis, such as how much fraction of the smaller mode, how will this fraction be related to the sources. Some very relevant references may be helpful to aid this observation (doi:10.5194/acp-14-10061-2014; doi:10.5194/acp-12-1681-2012).

Specific:

Fig.3 I would like to see a full set of rBC core size distribution for all of the DMA sizes, also the project standard deviation.

Fig.4 There is a significant fraction of tail on the rBC mass distribution. This seems to be two modes of BC distribution, maybe we could do a lognormal fitting on one mode and then the remaining is the other mode. and why is that?

Fig. 5 what do the small markers stand for? What is the point for the fitting?

Page 8-10, I found the whole section is a bit too lengthy but not really discussing your own results.

[Figure]

I found problematic for the fitting in Fig. 7. Because you are measuring the total particle mobility size but only the rBC mass content.

---

## Referee Comment (RC2) · Anonymous Referee #2 · 25 Jul 2016

It is very exciting to see SP2-measured rBC mass loadings and microphysical state information coming out of India. This important yet under-reported region is a critical piece in the "BC puzzle". Raatikainen et al,. present SP2 measurements of mobility-diameter selected aerosol, and have obtained data which, if appropriately quality-assured, will be of high value and interest to the wider community.

The manuscript needs substantial improvement. Most significantly, the quality of the rBC data set should be more carefully assessed: a ~20% bias to a CPC was corrected simply based on the assumption that the CPC was correct. Potentially larger biases than this exist in the data.

Generally, the manuscript should be proof-read for grammar. Imprecise vocabulary

(e.g. "absorbing particles" for rBC-containing particles) should be revised (there are many light-absorbing particles that may not contain rBC). Background information about the performance of the SP2 should be more clearly summarized in the main text. I also found it difficult to follow how the size-resolved results were combined to reflect the ambient aerosol condition. The results should be more thoughtfully presented (for example, the section about BC volume fractions based on the assumption that the total particle volume is proportional to mobility diameter cubed is followed by a section indicating the lack of value of that analysis after inspection of SP2 LEO fit results).

Specific comments:

1) "Absorbing" and "refractory" are terms that apply to rBC, but not all light-absorbing or refractory particles are either rBC-containing, or detected via incandescence in the SP2. For example, only a small fraction of dust particles incandescence in the SP2, although most of them are refractory. The paper should be made more precise by avoiding un-specific vocabulary. Two examples: 1) page 2 line 13 – the SP2 will not help determine a number fraction of light-absorbing particles; 2) page 3, line 19: the Sp2 does not measure all "refractory absorbing material". 2) Page 1 line 18: BC mixing state also depends on source. 3) Page 2 Line 11 and page 4, Line 17 and page 9 lines 5 - 11: The light-scattering for rBC-containing particles from the SP2 is a direct measurement (this is essentially a measurement of the particle optical size), and requires no assumptions about particle index of refraction or morphology. However, interpretation of the light scattering in terms of particle composition etc. does require assumptions. 4) Methods: a. Please specify if the aerosol was dried before sampling. b. It is necessary to consider SP2 detection efficiency of rBC, which depends on laser intensity and is influenced by mixing state (see Laborde et al., 2012 – the AIDA intercomparision in AMT, and Schwarz et al., 2010 – Detection efficiency of the SP2 in AMT). A first order estimate of laser intensity can likely be obtained from "YAG power" if Droplet Measurement Technology provided a calibration of this. Note that "YAG power" and laser intensity inside the chamber (where it is critical for rBC detection) are both temperature sensitive. The statement in the supplement that laser power changes equivalent to 17% in diameter suggests even larger laser intensity changes (depending on the size of the calibration aerosol but likely more than a factor 2; please present the relative change in laser intensity inferred from the ambient measurements and assuming constant index of refraction). c. The supplemental results showing dependencies on detection from the SP2 on chamber temperature (figure S4) for the DMA set at a quite large size (285nm) are quite concerning since the scattering-aerosol detection limit was specified to be 180 nm in the main text. The suggestion that the CPC is a reference instrument and the SP2 is of low quality is not sufficient to simply correct the SP2 concentration without testing any hypothesis as to the origin of the offset (CPC's can undercount!). Over the temperature range specified, the SP2 LFE and Ashcroft differential pressure meter have much smaller temperature sensitivities than the 30% shown in the figure. Hence the basis for a correction, and for establishing the absolute uncertainty of the SP2 measurement is not clear. Note that the good correlation of Aethalometer and Sp2 data is insufficient to rule out potentially large bias in either/both instruments, and does not validate the rBC size distributions, which have much smaller impact on rBC absorption than concentration; the Aethalometer result needs a large scaling factor that depends on total aerosol, and is highly site specific (see, for example: http://www.esrl.noaa.gov/psd/iasoa/node/81). It will be necessary for the authors to build a reasonable basis for evaluating the instrumental uncertainty. As part of this, I recommend including (in the supplemental material) peak-height distributions for the size-selected aerosol from the SP2 from these tests. d. Page 3, line 18: here the SP2 is indicated to detect both rBC-containing and rBC-free aerosol; this lead me to be confused about if the SP2-derived rBC-free concentration was used for anything other than comparison of detection efficiency with the CPC – was Sp2 rBC-concentration used with SP2 rBC-free concentration for the number fraction rBC-containing particls?. Can the data sources for the various parameters be included in a table or clearly summarized in the text? e. Please briefly summarize the inversion for the reader here in the main text. I found the supplemental material also confusing, as

the DMPS number size distribution comes from the DMA + the CPC: how is the ratio of "DMPS number size distribution to CPC concentration" particularly meaningful? The basic point is that SP2 concentrations are corrected to account for the number fraction of particles in a particular size bin with other than 1 electron charge? Please expand to explain how multiply charge particles are dealt with; this is likely a question that is in readers minds when looking at figure 3. This is at the heart of interpreting the "basic" SP2 data of rBC concentration and size distribution. f. Section 3.1: g. The Supplemental figure S1 should be updated to show that the Aethalometer sampled in parallel with the DMA. 5) Results: a. Page 4, line 25: What is the basis for this assumption? Every SP2 user sees rBC associated with singly and doubly charged (at least) peaks when calibrating with Aquadag or other materials (this is the bare rBC case). In the case of size-selected internally mixed particles, why would a narrow rBC size distribution be assumed? b. Section 32. Please address multiply-charged particles in the DMA. Page 5, line 18 – As these particles appear to be mostly bare, fractal rBC, how do you know that the ∼210 nm mode are not doubly-charged particles? This analysis would be helped by including the information from LEO fitting which was done. A prioriy one would expect that, unless there were dramatic changes in fractal dimension (which the authors rule out), a single size selected aerosol population will contain a continuum in which more massive rBC cores are associated with lesser amounts of non-rBC material, and smaller cores are associated with larger amounts of non-rBC. c. Page 6, line 1: note that rBC mass is a better first order proxy for the absorption cross-section in each size bin. d. Page 6, line 22 – 27: what are the values and total uncertainties observed here Page 6, line 30: Secondary rBC modes at larger sizes have been observed in China, these should be cited here: Wang et al., Shen, Black carbon aerosol characterization in a remote area of Qinghai–Tibetan Plateau, western China, Science of The Total Environment, V.479–480:151-158 (2014) and Huang et al., Black carbon measurements in the Pearl River Delta region of China, Journal of Geophysical Research, 116(D12208), doi:10.1029/2010JD014933, 2011. Is anything similar observed here? e. Page 7, line 6 and line 25: the volume ratio goes as the

ratio of diameters cubed, not as the linear ratio. This should be corrected throughout the paper. Line 21: The low ratio of rBC to total volume is in fact not distinctly different from previously published SP2 results based on LEO interpretation with Mie shell-and-core model. The assumption of this analysis (that Dm provides a route to total particle volume even for rBC-containing particles) is shown to be false in the next section; the results should be more clearly presented. f. Page 9 – why aren't any of the LEO results shown? What was the average scattering cross-sections? What fraction of rBC-containing particles were successfully fit? If the particles are bare, the scattering cross-section to rBC mass relationship should match expectations for material with the index of refraction measured by Moteki et al., Method to measure refractive indices of small non-spherical particles: Application to black carbon particles, J. Aero. Sci., 2010. The analysis of page 10 requires the LEO data, or analysis of the evolution of scattering signal for individual rBC particles, to support the assumption that the rBC particles are bare. 6) Conclusions: Please include quantitative values with uncertainties in this section. 7) Table 1: the $\pm$ values are standard deviations? This should be stated in the caption. Please include absolute uncertainties for rBC concentrations at least. 8) Figure 2: The time series does not seem to be extensively referenced in the text, perhaps this figure should be moved to supplemental material? 9) Figure 5: DMA data down to 20nm was taken. Why is the full range not shown if the clearly biased data below 200 nm is shown? I would prefer to see the whole range. Here it seems that the CPC data was not used for total particle number – why not? And if not, how is rBC number fraction ∼50% down to 100 nm when the SP2 scattering particle detection limit is specified to be 180 nm?

---

## Referee Comment (RC3) · G R McMeeking (Referee) · 31 Jul 2016

The manuscript presents measurements of refractory black carbon measurements collected at two sites in northern India using a relatively new approach featuring a single-particle BC instrument. By placing an SP2 downstream of a DMA the authors were able to examine relationships between mobility diameter and SP2 data products, such as refractory BC mass, non-rBC containing particle optical size, and derived rBC-containing particle size. Based on the measurement location and types of measurements conducted the manuscript has the potential to be a significant contribution, however requires substantial revision before I can recommend it for publication in ACP.

General Comments

[Figure]

The main weakness of the current manuscript is the inconsistent, incomplete and at times confusing treatment of the main, unique data product of the DMA-SP2 combination, the relationship between rBC and total particle "diameters". It is very important to highlight and appropriately account for the difference in the mobility diameter selected by the DMA, which depends on particle shape and physical size, and the mass of refractory material measured by the SP2. For a given selected mobility diameter rBC mass will depend on whether the particle is mixed with other materials and the effective density or shape of the rBC particle. At times the analysis assumes particles are spherical in order to compare rBC volume and total particle volume, but in other places the manuscript specifically states the particles are non-spherical aggregates. I think there is additional information (SP2 data products, ambient conditions) available to better identify situations where it is "safer" to treat the data in one extreme or the other (e.g., highly aged/spherical versus fresh/fractal). Reviewer #2 makes a number of very insightful comments regarding this aspect of the analysis, so I will not repeat many of her or his points here, however I would like to stress that this area of the analysis should be revisited, with particular attention paid to how uncertainties in the particle shape vs mixing affect the major conclusions of the paper.

There are many locations in the manuscript where the authors could be more specific without forcing the reader to consult tables and/or supplementary material. The methods section, not supplementary material, should give important details such as the scan time being 30 minutes or that the sample was dried (to what RH?). Was an impactor used upstream of the DMA to remove larger particles that would affect the inversion? In addition, it would be stronger if more specific numbers were used rather than more general terms like more, less, or different.

Specific comments (page-line)

1-18: I recommend the authors use a different term here than elemental carbon, which to most readers will be interpreted as the operationally defined measurement, when I think they simply mean to say that the BC particles are initially externally mixed and

"pure BC" for lack of a better term.

2-2: I suggest a slight re-phrasing, since being a CCN is not necessarily a requirement for cloud processing (e.g., collision processes)

2-10: "from" to "for"?

2-29: Avoid use of "truly", because "size" is not a well-defined term (diameter, equivalent diameter, mass, surface area?). Better to say "mobility diameter" as that is the property relevant to the DMA used here.

3-26: Suggest very briefly stating what is involved in the inversion without going into details placed in supplementary information to give reader a sense of what is being done here. For example, "Number concentrations measured by the SP2 were converted to rBC number distributions by accounting for charged particle fractions and the DMA transfer function using standard DMPS inversion methods (see Supplementary Material)". Note the inversion does matter for reporting averages like number fraction or volume fraction if they include all particles and not just those identified as +1 charged.

3-29: The manuscript should also cite our original HTDMA-SP2 methods paper here (McMeeking et al., 2011). I also recommend the authors mention other work where the SP2 has been placed behind different types of classifiers, such as the APM and/or CPMA, such as Ohata et al. (2016).

4-1/3: Another important thing to note here is higher output concentrations going through only one DMA instead of the two used in the VTDMA and HTDMA studies, allowing for improved counting statistics and faster scan times.

4-23: Please clarify: I think the authors are referring to the SP2 size resolution? They should also mention a range of output sizes would be expected due to the DMA transfer function and that would have to be accounted for. It would also be useful to cite an example from literature supporting the claim that most assume a narrow distribution of rBC core sizes...I'm not sure this is really the case.

Section 3.1 – it would be useful to have the average BC concentrations at each site given here, along with variability metric. Also better to state which of the sites was higher than saying there was a difference between them. Finally the measurements were performed at different times, so it would be helpful to know if there is any general trend in concentrations during the region over this time using the Aethalometer data.

5-7: I found this confusing. What is really meant by "homogenous" here. . .the authors mention multiple times in the paper that BC has different mixing states and morphologies, which could both affect the rBC vs DMA size relationship. Please elaborate.

5-18: It might be possible to determine whether the larger particles are indeed multiply charged particles based on the rough sizing metrics available from the SP2 itself (and could be further checked against known charging fractions).

7-3: "it is expected that particles are spherical" Can the authors support this statement? I'm not sure if they meant "assumed" instead of "expected", since they caveat the entire statement with "For simplicity, . . ." Is this section exploring a hypothetical exploration of the data, or real? Unless there is any strong evidence that particles are spherical at these sites, especially the rBC particles, than suggest re-phrasing to "For simplicity, we treat particles as spherical. . .". See general comments above.

7-13: Sulfate is not really considered semi-volatile under typically atmospheric conditions. . . Further, primary semi-volatile organics would, if anything, evaporate as concentrations decrease away from the urban areas leading to increases in the rBC volume fraction, though this could be counter-acted by formation of additional secondary semi-volatiles. I think "secondary" would be more accurate.

7-21: Are these numbers for total particles or just for the selected mobility diameter. Is the comparison to the literature values cited just before this statement describing the same (e.g., all or size-selected)?

7-23/24: Is there any evidence that BB is the dominant source? Biomass burning (depending on type, fuel, efficiency) will also emit non-rBC containing particles. Could other sources still be important? These might emit higher number fractions of rBC-containing particles compared to biomass burning. Were there any patterns in week-day versus weekend traffic or activities in these regions that might affect the sources?

8-15/16: I agree with the vertical mixing affecting aerosol concentrations, but don't see how dilution would affect the rBC number fraction, which would be independent of concentration, unless some aerosol micro-physical processes are going on. Is there evidence that the aerosol outside the nocturnal boundary layer has lower rBC number fractions and mixes into the surface layer?

8-21: It is not clear whether the authors are comparing the two sites here, or the periods when the Gual Pahari or polluted Indo-Gangetic plains particles affect the site to those when there is little impact?

9-3: Please be more specific about what is meant here by the phrase "current experiments"...is this recent work in the field or specific to the Indian measurements?

9-9/11: I think this particular section represents the general problems with how the treatment of the uncertainties in the different size classification methods affects the conclusions of the paper. How was the cutoff of the 10% difference determined to know a particle was non-spherical? Does it account for an expected range of rBC and non-rBC refractive indices, which will also affect the difference between optical and mobility diameters? The authors should also provide the uncertainty in the LEO method for their instrument based on a comparison of the full Gaussian scattering measurements and LEO results for non-rBC containing results to support criteria further.

9-31/10-20: Unless I have missed something I believe the analysis described here treats the rBC particles as purely rBC particles to calculate the mass fractal dimensions? This contradicts earlier discussions in the paper regarding volume fraction. It would be better to restrict the calculations to particles likely to have less other material (e.g., smaller LEO size, maximum rBC mass observed in each DMA mobility size). I

don't see an easy way to distinguish the shape effect on DMA sizing from the effects of other materials mixed with the rBC if this is not done.

Figure 4: Remove the spurious low point that is related to the gain-stage matching in the SP2. Better to show as a gap than as a non-realistic lower data point.

---

## Short Comment (SC1) · 10 Aug 2016

I have three short comments:

1. Page 1, Line 20: ". . ., but this also depends on the structure of the particle." Here, the authors missed several more recent studies on the morphological effects on BC optical properties. For example, He et al. (2015, 2016) showed that different structures for both fresh and coated BC particles can lead to substantial variations in BC absorption and scattering. I suggest including these two studies as references here.

References:

He, C., Liou, K.-N., Takano, Y., Zhang, R., Levy Zamora, M., Yang, P., Li, Q., and

Leung, L. R.: Variation of the radiative properties during black carbon aging: theoretical and experimental intercomparison, Atmos. Chem. Phys., 15, 11967-11980, doi:10.5194/acp-15-11967-2015, 2015.

He, C., Takano, Y., Liou, K.-N., Yang, P., Li, Q., and Mackowski, D. W.: Intercomparison of the GOS approach, superposition T-matrix method, and laboratory measurements for black carbon optical properties during aging, J. Quant. Spectrosc. Radiat. Transfer, in press, 2016.

2. Page 2, Line 17: ". . ., anthropogenic emissions such as biomass burning . . ." Typically, biomass burning is referred to wildfire emission, which is not included in the anthropogenic (i.e. fossil fuel and biofuel burning) emissions. Here, are you referring to agricultural burning? Please clarify.

3. Page 9, Lines 12–25: as stated in my first comment, recent studies (He et al., 2015, 2016) have shown that nonspherical/fractal structures of both fresh and coated BC particles can significantly affect BC optical properties and hence optical size during measurement. This could introduce large uncertainty into the optical method. The authors also showed results suggesting highly fractal BC aggregates in the measurement. In addition, the authors assumed the core-shell structure to quantify BC mixing state, which could also bring some uncertainty to their results due to the irregular coating structures in the real atmosphere. Thus, I suggest adding some discussions on how fractal aggregating structures could possibly increase the uncertainty in measurements and analysis.

---

## Author Comment (AC1) · 25 Oct 2016

We would like to thank Referee #1 for the constructive comments. The comments below are shown with italicized font and our replies with the upright font. The changes to the manuscript and Supplementary Material are specified in the supplement.

*I think this paper needs to be more conclusive (too many hypothesis at the moment) before it can be accepted. The writing itself I feel a bit lengthy. The major points in particular:*

*-the BC sources are not clear, some back trajectory analysis will be helpful. This unclearness goes through all of the texts when discussing if the source is local or*

[Figure]

*transported etc. The discussions on diurnal variation are also weak because of lack of source analysis. With clear source analysis, these discussions should be tidier.*

We have done back trajectory analysis, but unfortunately we did not see clear correlation between our rBC observations and trajectory parameters such as average direction or altitude. Therefore, the trajectory analysis was left out from the manuscript. We have now shortened the discussion about sources, because we don't have any additional information about these.

*-the main limitation is most of the information is derived from DMA360nm, which could only represent a fraction of total BC. The most populated total particle size may not represent the most populated rBC size, therefore combing the rBC size at other DMA sizes will be also useful. I would suggest to analyse and compare the rBC information at all DMA sizes together.*

rBC mass and number size distributions (described by the log-normal distribution parameters) are derived using the available DMA sizes ($>$ 100 nm). Size-resolved results such as the rBC core mean diameters are represented by the bin mean values (Fig. 5) and time series from one DMA size (360 nm). Only one DMA size is selected, because the parameters are so similar that selecting another DMA size would lead us to the same conclusions. Better explanation for selecting only the 360 nm DMA size is given in the updated manuscript.

*-the chamber temperature introduces some instrument bias, I guess the SP2 laser power was affected by this? how have you corrected this regarding the reduced detection efficiency when reduced laser power?*

We have added laser power diagnostic to the Supplementary Material. It seems that laser power has decreased in Gual Pahari so that the detection efficiency is lower for rBC cores when particle size is smaller than 220 nm. A correction would have required detailed quantification of the detection efficiency, which is not possible anymore. We

have therefore limited our calculations to sizes larger than 200 nm, where the effect of the detection limit seem to be small, and used the much larger 360 nm mobility size bin in most calculations.

*-the BC mixing state as derived from DMPS-SP2 is not clear, did you calculate as rBC size divided by mobility size? I don't think page 7 has explained what you have done sufficiently. This is really important but this only appears in supplement. The uncertainty of this method is largely from the particle morphology, however it is hard to tell without particle mass measurement (though you measured rBC mass but this is not the total BC-containing particle mass). I think the main texts need to address this uncertainty aided by more thorough analysis at different DMA sizes.*

The text in page 7 was indeed unclear regarding the calculations, but it has now been revised. We will also clarify that both rBC and mobility diameters are measured quantities and therefore accurate within the typical measurement uncertainties. We no longer assume that the particles would be spherical, but just use the measured parameters. The reasons for focusing on the single DMA size are given above and in the updated manuscript.

*-The bimodal mode of rBC under cleaner environment looks interesting which needs more detailed analysis, such as how much fraction of the smaller mode, how will this fraction be related to the sources. Some very relevant references may be helpful to aid this observation (doi:10.5194/acp-14-10061-2014; doi:10.5194/acp-12-1681-2012).*

The fraction of the larger mode is given in the text (0.5-0.8) and the rest are from the smaller mode (0.2-0.5). The two references are familiar to us, but their source analysis is based on different instruments and numerical methods (positive matrix factorization of high-resolution aerosol mass spectrometer data in doi:10.5194/acp-14-10061-2014 and cluster analysis of aerosol time-of-flight mass spectrometer data in doi:10.5194/acp-12-1681-2012). These methods cannot be applied to the current

data, because the different DMA sizes are strongly correlated (see Fig. 5). We have done source analysis using trajectory data, but specific sources could not be identified. Therefore, the source analysis was left out of the manuscript.

**Specific:**

*Fig.3 I would like to see a full set of rBC core size distribution for all of the DMA sizes, also the project standard deviation.*

All DMA sizes do not provide valid information and the relevant DMA sizes are correlated, so that the same conclusions could be made based on any of these. However, the Supplementary Material has been updated with the rBC core size distributions for the usable DMA size range and also the standard deviations are shown. Because standard deviations make the figures less readable, these are shown only in the Supplementary Material.

*Fig.4 There is a significant fraction of tail on the rBC mass distribution. This seems to be two modes of BC distribution, maybe we could do a lognormal fitting on one mode and then the remaining is the other mode. and why is that?*

Bimodal log-normal fits have been added to Fig. 4, but only the smaller mode is fully resolved due to the SP2 sizing limits. Fractions of the modes in the SP2 sizing range are now reported. We don't have good explanation for the origin of the large particles.

*Fig. 5 what do the small markers stand for? What is the point for the fitting?*

Small markers indicate the data that is not used in the fitting. It is now clarified these data points are ignored (due to instrument detection limits or poor counting statistics). The fits are convenient parameterizations for the rBC mixing state parameters, which can be used in other studies.

*Page 8-10, I found the whole section is a bit too lengthy but not really discussing your own results.*

This section has been made significantly shorter and more focused on our results (especially LEO). Especially the text about mass fractal dimension parameter has been removed completely.

*I found problematic for the fitting in Fig. 7. Because you are measuring the total particle mobility size but only the rBC mass content.*

This figure has been removed (see the previous comment).

Please also note the supplement to this comment:
http://www.atmos-chem-phys-discuss.net/acp-2016-435/acp-2016-435-AC1-supplement.pdf

———————————————————

[Figure]

**Supplement:**

[revised manuscript text omitted]
 (first drier (using silica-gel, Sigma Aldrich) was in main inlet line outside, second stage was 1 meter-long nafion drier (Perma Pure) inside the building, third drier (silica-gel, Sigma Aldrich) was in closed DMA sheath flow loop) and the charge distribution is neutralized before entering the DMA where a narrow particle size range is selected based on a voltage control (VDC) while keeping the constant flow rates (both sheath and aerosol sample: $Q_a$). The size-selected aerosol is then lead to the CPC, which measured the total number concentration of particles ($N_{total}$). The SP2 was connected in parallel to the CPC where the SP2 measures the number concentrations of all particles ($N_{total}$) and also those containing refractory material ($N_{rBC}$), which is assumed to be refractory black carbon (rBC). In addition, the SP2 measures particle sizes and the mass of refractory material ($m_{rBC}$) in each particle. Particle size measured by the SP2 is  used mainly as a diagnostic parameter; for spherical particles it should be the same as the mobility size selected by the DMA, which is used in the calculations. rBC core volume equivalent diameters (assuming a compact spherical rBC "core" with 1800 kg m$^{-3}$ density) were calculated from the measured rBC mass.

[Figure]

Figure S1: Schematics of the measurement setup.

The DMA voltages are changed stepwise through selected voltage range. Here 30 voltage settings were selected so that logarithmically spaced particle size bins ranged from about 20 to 650 nm (there were small fluctuations due to instrument temperature and pressure changes). The time step was initially set to 30 s, but it was later increased to 60 s to improve the SP2 counting statistics. As a result, the full scan with 60 s time step took 32 minutes.

The CPC and SP2 data were aligned visually. The CPC data is automatically saved as an average concentration for each time step, but this was done manually for the SP2 data where the relevant parameters are number concentrations for all particles and those containing rBC (used to calculate the number fraction of particles containing rBC), and rBC core size distributions which are here represented by suitable distribution parameters. These were calculated using the standard instrument calibration and data analysis methods (see below). The first measurements showed that the rBC core size distributions are not always narrow and unimodal, which means that the commonly used number mean rBC core diameter is

smaller than the corresponding mass mean diameter. For consistency with the further mass calculations, both mass and number mean diameters were saved.

**1.1 SP2 calibration**

The maximum incandescence and scattering signals are proportional to the refractory particle mass and the total particle size, respectively, and these dependencies are parameterized using suitable calibration reference materials. Our regular incandescence calibrations are based on size-selected Aquadag (Acheson Inc., USA) particles which masses are calculated using the density parameterization from Gysel et al. (2011). Several studies have shown that fullerene soot is the best representative of ambient rBC and its incandescence signal is about 75% of that of Aquadag with the same particle mass (Moteki and Kondo, 2010; Kondo et al., 2011; Gysel et al., 2011; Baumgardner et al., 2012; Laborde et al., 2012). Therefore, Aquadag incandescence amplitudes were multiplied by a factor of 0.75 to make it representative of the fullerene soot and ambient rBC. This fullerene soot-equivalent calibration gives ambient rBC particle mass as a function of the measured maximum incandescence signal. rBC core volume equivalent diameters (briefly: rBC core diameter)  are then calculated by assuming a compact spherical "core" with 1800 kg m$^{-3}$ density. The sizing limits for the refractory material are 0.3-380 fg for rBC mass and 70-740 nm for the rBC core diameter.

Because particle sizes are accurately selected by the DMA especially for rBC-containing particles, scattering sizes measured by the SP2 are mainly used for diagnostic purposes. Although we do not use scattering sizes measured by the SP2 (these are accurately selected by the DMA especially for absorbing particles), we calculate those for diagnostic purposes. Our regular scattering size calibrations are based on size-selected ammonium sulfate, but here we can also use size-selected ambient particles. The instrument was calibrated using ammonium sulfate before shipping to India (December 13, 2013), but not during the campaign. Scattering amplitudes from the ammonium sulfate calibration and Mukteshwar observations (February 9, 2014) are almost identical with only 3% difference in particle size. Such a small difference can be easily caused by the difference in optical properties of ammonium sulfate and ambient particles or minor changes in the instrument performance. However, significant reduction in scattering signal (equivalent to 17% increase in particle size) is observed after the instrument was moved to Gual Pahari (data from April 11, 2014). This is most likely caused by decreased laser beam intensity (Laborde et al., 2012). In order to remove this constant sizing bias, Gual Pahari scattering calibration is based on field measurements (data from April 11, 2014). More details about the laser power is given in later in Sect. 3.4.

**2 Data analysis**

Just as in a typical DMPS data analysis, ambient size distributions need to be calculated from the raw observations of particle number versus voltage using an inversion code. This inversion depends mainly on particle charging efficiencies and the DMA transfer function. For the DMPS system we have used a standard pseudo-inverse method (see the FMI DMPS inversion description in Wiedensohler et al., 2012), but this method cannot be directly applied to the SP2 data. The main reason is that the SP2 cannot detect small particles (minimum rBC size is about 70 nm and that for scattering particles is roughly 180 nm), which means that multiply charged particles dominate these size bins a large fraction of the size scan particle counts are originating from multiply charged particles. To avoid problems with the data inversion, SP2 size distributions (dN$_{SP2}$/dlogD$_p$) concentrations are calculated by multiplying the raw particle number concentrations (N$_{SP2}$) by the ratio of the inverted DMPS number size distribution (dN$_{DMPS}$/dlogD$_p$) to the original using the inversion results particle concentrations from the CPC (N$_{CPC}$): dN$_{SP2}$/dlogD$_p$=

$N_{SP2}\cdot(dN_{DMPS}/dlogD_p)/N_{CPC}$ In order to reduce noise from the inversion and to make SP2 results less dependent on the CPC data, we have used the modethe correction factor (($dN_{DMPS}/dlogD_p)/N_{CPC}$), which means constant correction for each DMA size. This method is essentially the same as taking a typical DMPS scan and calculating the inversion from that. Figure S2 shows an example of the SP2 inversion method. The red markers are the individual time and size dependent values of ($dN_{DMPS}/dlogD_p)/N_{CPC}$ and blue lines and markers represent the mode, which is the size dependent correction factor  for the SP2 concentrations (here number concentrations of particles with and without rBC, rBC mass concentrations, and rBC core number size distributions for the 360 nm mobility size bin).

Parameters calculated for each DMA size like the  rBC core diameter,  scattering particle size and number fraction of rBC-containing  particles are independent of particle concentrations, so inversion is not needed. Multiply charged particles have a small effect on rBC core diameters (see Fig. S14), but the exact quantification of their contribution or a correction would have required accurate identification of those particles, which is not possible.

[revised manuscript text omitted]

**3.4 Laser power diagnostics**

Figure S9 shows the development of the low gain scattering signal amplitudes for the 285 nm DMA size during the whole measurement period (blue and red marker colors represent Mukteshwar and Gual Pahari, respectively). Also shown is the chamber temperature, which can explain most fluctuations in scattering amplitudes (and particle sizes as shown above). The normalization is based on scattering calibration made just before sending the instrument to India (13.12.2013). On the average, the scattering signals in Mukteshwar and Gual Pahari are reduced to 93% and 41% from the calibration level. The reason for the decrease in the scattering signal when the instrument was transported from Mukteshwar to Gual Pahari is unknown, but it is possible that laser power has decreased.

Too low laser power could mean that rBC in the smallest and/or thickly coated particles is not detected. This is examined by calculating the positions for the maximum incandescence signal. The maximum incandescence signal should be seen before particles cross the laser beam center. Figure S10 shows the distributions of times from incandescence signal to the laser beam center for different mobility diameters and rBC core diameters (half of the mobility diameter ±5 nm). Variable core sizes were selected so that these would be representative of the typical particles. The data has been selected from one day with low scattering signal amplitudes. Figure S10 shows that laser power is high enough for particles with mobility diameter at least 220 nm (rBC core diameter 110 nm). Smaller mobility diameters and rBC cores may have reduced detection efficiency especially in Gual Pahari. The 200 nm mobility size limit used in the main text is therefore suitable for rBC.

[Figure]

Figure S9: SP2 chamber temperature and measured and normalized scattering amplitudes for Mukteshwar (blue markers) and Gual Pahari (red markers). Scattering amplitudes are for the 285 nm DMA size. The normalization is based on scattering calibration made before sending the instrument to India (13.12.2013).

[Figure]

Figure S10: Normalized distributions of incandescence to laser beam center time bins for Gual Pahari and Mukteshwar. The distributions are shown for different mobility and rBC core diameters (core diameter is half of the mobility diameter).

Significant problems with laser could also affect scattering peak height distributions, but such problems were not seen. As an example, scattering peak height distribution for the 285 nm mobility size bin is shown in Fig. S11 for both Gual Pahari (top) and Mukteshwar (bottom). The maximum peak height for the current instrument is about 65000 (depends on the baseline noise) and the peak close to this limit corresponds to saturated signal. As is already shown in Fig. S9, scattering amplitudes are somewhat lower in Gual Pahari.

[Figure]

[Figure]

Figure S11: Scattering peak height distributions for 280 nm mobility size for Gual Pahari (top) and Mukteshwar (bottom).

**4 Supplementary figures**

In addition to the average rBC mass size distributions (total mass and geometric mass mean diameter and standard deviation) and mixing state (number and volume mean core diameters, those normalized by the mobility size, and number fraction of particles containing rBC) parameters for the 360 nm mobility size bins given in the main text (Table 1), figures S10 and S11 show the time series of these parameters (except number and volume mean core diameters as these are proportional to the normalized values).

[Figure]

Figure S10̶2: Time series of the measured rBC mass size distribution parameters (total rBC concentration and geometric mass mean diameter and standard deviation) and mixing state parameters (rBC core volume equivalent diameter (both number and volume averages) to mobility̶particle diameter ratio and number fraction of particles containing rBC) for Mukteshwar.

[Figure]

Figure S11̶3: Same as Fig. S10̶2, but for Gual Pahari

Figure 3 in the main text shows the campaign average rBC core number size distributions for the 360 nm DMA-selected mobility diameter. Here Fig. S14 shows the core number size distributions for 220 nm, 360 nm and 520 nm mobility diameters and also with one standard deviation error bars. The 360 nm figure is the same as in the main text, but now with the error bars, and the other two sizes represent the lower and upper limits for accurate measurements. As mentioned in the main text, the mode with smaller rBC cores observed in Mukteshwar is below the detection limit (70 nm) for the measurements with 220 nm mobility diameter. The error bars represent the variability of the rBC mass concentrations.

[Figure]

Figure S14: Campaign average rBC core number size distributions for 220, 360 and 520 nm DMA-selected mobility diameters (indicated by the vertical gray lines) from Mukteshwar (multiplied by a factor of ten) and Gual Pahari.

Refractory black carbon size mass and number distributions as a function of rBC core volume equivalent diameter are shown in the main text (Fig. 4). Figure S15 shows rBC and total particle number and mass size distributions measured by the SP2 and DMPS as a function of the DMA-selected mobility diameter.

[Figure]

Figure S15: Total and rBC mass and number size distributions measured by the DMPS and SP2 for Gual Pahari and Mukteshwar. The volume distributions measured by the DMPS have been converted to mass by using 1000 kg m$^{-3}$ particle density.

LEO analysis was conducted as described in the main text. Time series of the particles sizes (number average for each hour and mobility size bin) were calculated for different particle types (with and without rBC). Although the LEO method seems to work well for particles without rBC, examination of the chamber temperature dependency reveals problems with rBC-containing particles. Figure S16 shows the dependency of the LEO-derived optical particle size on the SP2 chamber temperature for particles without (top) and with rBC (bottom) for the 360 nm mobility size bin. When the size for non-rBC particles is independent if the instrument temperature, this is not the case for rBC-containing particles. The exact reason for this dependency is unknown, but an unusual sinusoidal noise signal seen on top of the Gaussian scattering signal seem to disturb the LEO fit especially at chamber temperature below 30 °C. For this reason, LEO results are presented briefly in the main text (only averages for the 360 nm mobility size and when chamber temperature is at least 30 °C).

[Figure]

Figure S16: LEO-derived optical particle sizes as a function of chamber temperature for the 360 nm mobility size bin. The top graph is for particles without rBC and the bottom graph is for rBC-containing particles. Clear outliers are not shown.

[revised manuscript text omitted]

---

## Author Comment (AC2) · 25 Oct 2016

We would like to thank Referee #2 for the constructive comments. The comments below are shown with italicized font and our replies with the upright font. The changes to the manuscript and Supplementary Material are given as a supplement with replies to Referee #1.

*It is very exciting to see SP2-measured rBC mass loadings and microphysical state information coming out of India. This important yet under-reported region is a critical piece in the "BC puzzle". Raatikainen et al,. present SP2 measurements of mobility-diameter selected aerosol, and have obtained data which, if appropriately quality-*

[Figure]

*assured, will be of high value and interest to the wider community.*

*The manuscript needs substantial improvement. Most significantly, the quality of the rBC data set should be more carefully assessed: a ~20% bias to a CPC was corrected simply based on the assumption that the CPC was correct. Potentially larger biases than this exist in the data.*

We have now confirmed that this 20% bias is systematic and related to this specific SP2. The counting was verified by our other SP2-D, which shows the same concentration as CPC used in Indian campaign and as other calibrated CPCs. The over counting is size independent; however, the reason for this over counting is still unknown.

*Generally, the manuscript should be proof-read for grammar. Imprecise vocabulary (e.g. "absorbing particles" for rBC-containing particles) should be revised (there are many light-absorbing particles that may not contain rBC). Background information about the performance of the SP2 should be more clearly summarized in the main text. I also found it difficult to follow how the size-resolved results were combined to reflect the ambient aerosol condition. The results should be more thoughtfully presented (for example, the section about BC volume fractions based on the assumption that the total particle volume is proportional to mobility diameter cubed is followed by a section indicating the lack of value of that analysis after inspection of SP2 LEO fit results).*

We have improved the text and clarified the vocabulary. We have also clarified our terminology regarding the results. For example, volume fractions are not used anymore.

**Specific comments:**

*1) "Absorbing" and "refractory" are terms that apply to rBC, but not all light-absorbing or refractory particles are either rBC-containing, or detected via incandescence in the SP2. For example, only a small fraction of dust particles incandescence in the SP2,*

*although most of them are refractory. The paper should be made more precise by avoiding un-specific vocabulary. Two examples: 1) page 2 line 13 – the SP2 will not help determine a number fraction of light-absorbing particles; 2) page 3, line 19: the Sp2 does not measure all "refractory absorbing material".*

We have clarified this so that term rBC-containing is now used to describe those particles that are detected by the incandescence signal.

*2) Page 1 line 18: BC mixing state also depends on source.*

This is now clarified.

*3) Page 2 Line 11 and page 4, Line 17 and page 9 lines 5 - 11: The light-scattering for rBC-containing particles from the SP2 is a direct measurement (this is essentially a measurement of the particle optical size), and requires no assumptions about particle index of refraction or morphology. However, interpretation of the light scattering in terms of particle composition etc. does require assumptions.*

It is true that light scattering is measured directly, but these three parts of the text refer to particle size, which require assumptions.

*4) Methods:*

*a. Please specify if the aerosol was dried before sampling.*

It is now specified that the aerosol was dried (RH inside the DMA was about 25%). The ambient aerosol was dried in several stages, first drier (using silica-gel, Sigma Aldrich) was in main inlet line outside, second stage was 1 meter-long nafion drier (Perma Pure) inside the building, third drier (silica-gel, Sigma Aldrich) was in closed DMA sheath flow loop.

*b. It is necessary to consider SP2 detection efficiency of rBC, which depends on laser*

*intensity and is influenced by mixing state (see Laborde et al., 2012 – the AIDA intercomparision in AMT, and Schwarz et al., 2010 – Detection efficiency of the SP2 in AMT). A first order estimate of laser intensity can likely be obtained from "YAG power" if Droplet Measurement Technology provided a calibration of this. Note that "YAG power" and laser intensity inside the chamber (where it is critical for rBC detection) are both temperature sensitive. The statement in the supplement that laser power changes equivalent to 17% in diameter suggests even larger laser intensity changes (depending on the size of the calibration aerosol but likely more than a factor 2; please present the relative change in laser intensity inferred from the ambient measurements and assuming constant index of refraction).*

We have added diagnostics about laser power to the supplementary material. Scattering signal amplitude for about 280 nm particles was 93% at Mukteshwar and 41% at Gual Pahari from the amplitudes obtained in the initial laboratory calibration. We have also examined the locations of the incandescence signals, which should be seen before particles cross the laser beam center. These show that the typical rBC-containing particles from mobility diameters above 220 nm reach their incandescence temperatures before the maximum laser power, which indicates sufficient laser power. Laser power was higher in Mukteshwar, so a lower detection limit can be used.

*c. The supplemental results showing dependencies on detection from the SP2 on chamber temperature (figure S4) for the DMA set at a quite large size (285nm) are quite concerning since the scattering-aerosol detection limit was specified to be 180 nm in the main text. The suggestion that the CPC is a reference instrument and the SP2 is of low quality is not sufficient to simply correct the SP2 concentration without testing any hypothesis as to the origin of the offset (CPC's can undercount!). Over the temperature range specified, the SP2 LFE and Ashcroft differential pressure meter have much smaller temperature sensitivities than the 30% shown in the figure. Hence the basis for a correction, and for establishing the absolute uncertainty of the SP2 measurement is not clear. Note that the good correlation of Aethalometer and*

*Sp2 data is insufficient to rule out potentially large bias in either/both instruments, and does not validate the rBC size distributions, which have much smaller impact on rBC absorption than concentration; the Aethalometer result needs a large scaling factor that depends on total aerosol, and is highly site specific (see, for example: http://www.esrl.noaa.gov/psd/iasoa/node/81). It will be necessary for the authors to build a reasonable basis for evaluating the instrumental uncertainty. As part of this, I recommend including (in the supplemental material) peak-height distributions for the size-selected aerosol from the SP2 from these tests.*

We don't think that the temperature dependency shown in Fig. S4 is related to detection efficiency. If this would be the reason, then we would expect to see an increase in the detection efficiency with particle size (Fig. S3), but this is not the case. Hypotheses were tested, but we could not find good explanation for the difference between CPC and SP2. For example, flow rates were calibrated for both instruments before the campaign, so these should have been correct (this explanation has now been removed). Also, both instruments seem to be operating correctly and concentrations were low enough for the CPC to avoid coincidence effects. Additional tests with another CPC and SP2 showed that this specific SP2 is always over counting, so the correction is justified. This is now clarified.

It is true that the absolute values of the eBC are uncertain and eBC is not the same as rBC, so now we focus on the correlation. Also, the lack of temperature dependency of the eBC/rBC ratio indicates that the measured rBC is not temperature dependent. Peak height distributions are now shown in the supplement.

*d. Page 3, line 18: here the SP2 is indicated to detect both rBC-containing and rBC-free aerosol; this lead me to be confused about if the SP2-derived rBC-free concentration was used for anything other than comparison of detection efficiency with the CPC – was Sp2 rBC-concentration used with SP2 rBC-free concentration for the number fraction rBCcontaining particls?. Can the data sources for the various parameters be included*

[Figure]

*in a table or clearly summarized in the text?*

This section is now clarified. SP2-derived number concentrations are used in all calculations, and CPC is for diagnostics and inversion.

*e. Please briefly summarize the inversion for the reader here in the main text. I found the supplemental material also confusing, as the DMPS number size distribution comes from the DMA + the CPC: how is the ratio of "DMPS number size distribution to CPC concentration" particularly meaningful? The basic point is that SP2 concentrations are corrected to account for the number fraction of particles in a particular size bin with other than 1 electron charge? Please expand to explain how multiply charge particles are dealt with; this is likely a question that is in readers minds when looking at figure 3. This is at the heart of interpreting the "basic" SP2 data of rBC concentration and size distribution.*

Brief summary of the inversion is added and the description of the inversion in Supplementary material has been clarified. The SP2 concentrations are calculated using the DMPS inversion results, which account for the multiply charged particles. We have also clarified that multiply charged particles have a negligible effect (can be seen from Fig. 3).

*f. Section 3.1:*

*g. The Supplemental figure S1 should be updated to show that the Aethalometer sampled in parallel with the DMA.*

Figure S1 describes the DMPS-SP2 measurement system, which does not include the Aethalometer (it is another independent instrument and only used in Mukteshwar).

*5) Results:*

*a. Page 4, line 25: What is the basis for this assumption? Every SP2 user sees rBC associated with singly and doubly charged (at least) peaks when calibrating with Aquadag or other materials (this is the bare rBC case). In the case of size-selected internally mixed particles, why would a narrow rBC size distribution be assumed?*

This comment has been removed and a better explanation is given in Sect. 3.2.

*b. Section 32. Please address multiply-charged particles in the DMA. Page 5, line 18 – As these particles appear to be mostly bare, fractal rBC, how do you know that the ~210 nm mode are not doubly-charged particles? This analysis would be helped by including the information from LEO fitting which was done. A prioriy one would expect that, unless there were dramatic changes in fractal dimension (which the authors rule out), a single size selected aerosol population will contain a continuum in which more massive rBC cores are associated with lesser amounts of non-rBC material, and smaller cores are associated with larger amounts of non-rBC.*

The ~210 nm mode is always the dominant mode (also for larger mobility diameters), so it must be related to singly-charged particles. Any other conclusion would have required unrealistic rBC size distributions, which were not observed. We clarify that particles observed close to 360 nm are more likely from doubly charged particles than from pure compact rBC, and all larger particles have multiple charges. Nevertheless, these particles are a minor fraction, so they have a negligible contribution to the mean values. We have now provided more information about the LEO fit, but the analysis is kept limited due the problems with increased instrument noise (all LEO results given in Sect 3.4).

*c. Page 6, line 1: note that rBC mass is a better first order proxy for the absorption cross-section in each size bin.*

rBC mass mean diameter has been chosen, because it is familiar for typical readers (at least for us).

*d. Page 6, line 22 – 27: what are the values and total uncertainties observed here*

This paragraph has been removed, because rBC and eBC are not directly comparable.

*Page 6, line 30: Secondary rBC modes at larger sizes have been observed in China, these should be cited here: Wang et al., Shen, Black carbon aerosol characterization in a remote area of Qinghai–Tibetan Plateau, western China, Science of The Total Environment, V.479–480:151-158 (2014) and Huang et al., Black carbon measurements in the Pearl River Delta region of China, Journal of Geophysical Research, 116(D12208), doi:10.1029/2010JD014933, 2011. Is anything similar observed here?*

Reference to the Wang et al. (2014) has been added (Huang et al., 2011, was already included). We also clarify that similar larger particles were observed in these sites.

*e. Page 7, line 6 and line 25: the volume ratio goes as the ratio of diameters cubed, not as the linear ratio. This should be corrected throughout the paper. Line 21: The low ratio of rBC to total volume is in fact not distinctly different from previously published SP2 results based on LEO interpretation with Mie shell-and-core model. The assumption of this analysis (that Dm provides a route to total particle volume even for rBC-containing particles) is shown to be false in the next section; the results should be more clearly presented.*

Correction to the volume fraction has been made. We will also clarify that the numbers in line 21 are for rBC-containing particle number fractions and not rBC volume fractions. The ratio of rBC mean diameter to the mobility size is a directly measured quantity, although its interpretation is not as simple as for spherical particles. We have clarified the text so that diameter ratios are no longer interpreted as rBC volume fractions.

*f. Page 9 – why aren't any of the LEO results shown? What was the average scattering cross-sections? What fraction of rBC-containing particles were successfully fit? If*

*the particles are bare, the scattering cross-section to rBC mass relationship should match expectations for material with the index of refraction measured by Moteki et al., Method to measure refractive indices of small non-spherical particles: Application to black carbon particles, J. Aero. Sci., 2010. The analysis of page 10 requires the LEO data, or analysis of the evolution of scattering signal for individual rBC particles, to support the assumption that the rBC particles are bare.*

More details about the LEO fit are given and some additional LEO results are shown, but the discussion is kept limited due to the problems with the LEO fits. Note that we have not seen bare rBC, but irregular or fractal rBC. We have decided to report particle sizes instead of scattering cross sections, because particle sizes are more familiar for typical readers (at least for us).

*6) Conclusions: Please include quantitative values with uncertainties in this section.*

Values and uncertainties are now given.

*7) Table 1: the $\pm$ values are standard deviations? This should be stated in the caption. Please include absolute uncertainties for rBC concentrations at least.*

The values are standard deviations. Typical absolute uncertainty (20%) is given for rBC concentration.

*8) Figure 2: The time series does not seem to be extensively referenced in the text, perhaps this figure should be moved to supplemental material?*

This figure was originally in the supplementary material, but it was moved here by the request of the Editor, so we will keep it here.

*9) Figure 5: DMA data down to 20nm was taken. Why is the full range not shown if the clearly biased data below 200 nm is shown? I would prefer to see the whole range.*

[Figure]

*Here it seems that the CPC data was not used for total particle number – why not?
And if not, how is rBC number fraction ∼50% down to 100 nm when the SP2 scattering
particle detection limit is specified to be 180 nm?*

Figure 5 and all SP2 calculations were limited to the 100 nm mobility size, because this
is about the same as the SP2 detection limit for rBC, so the whole available range is
already shown. Mobility sizes above 100 nm range were fully analyzed and the results
were used to find the lowest size where results are still valid. Figure 5 shows very
clearly why the lower limit has been set to approx. 200 nm. CPC data was not used,
because concentrations are measured by the SP2. Any non-rBC particle count below
180 nm mobility diameter is related to multiply charged particles. Also, at 100 nm only
a fraction of rBC is detectable.

---

## Author Comment (AC3) · 25 Oct 2016

We would like to thank Referee #3 for the constructive comments. The comments below are shown with italicized font and our replies with the upright font. The changes to the manuscript and Supplementary Material are given as a supplement with replies to Referee #1.

*The manuscript presents measurements of refractory black carbon measurements collected at two sites in northern India using a relatively new approach featuring a single-particle BC instrument. By placing an SP2 downstream of a DMA the authors were able to examine relationships between mobility diameter and SP2 data prod-*

[Figure]

*ucts, such as refractory BC mass, non-rBC containing particle optical size, and derived rBC-containing particle size. Based on the measurement location and types of measurements conducted the manuscript has the potential to be a significant contribution, however requires substantial revision before I can recommend it for publication in ACP.*

*General Comments*

*The main weakness of the current manuscript is the inconsistent, incomplete and at times confusing treatment of the main, unique data product of the DMA-SP2 combination, the relationship between rBC and total particle "diameters". It is very important to highlight and appropriately account for the difference in the mobility diameter selected by the DMA, which depends on particle shape and physical size, and the mass of refractory material measured by the SP2. For a given selected mobility diameter rBC mass will depend on whether the particle is mixed with other materials and the effective density or shape of the rBC particle. At times the analysis assumes particles are spherical in order to compare rBC volume and total particle volume, but in other places the manuscript specifically states the particles are non-spherical aggregates. I think there is additional information (SP2 data products, ambient conditions) available to better identify situations where it is "safer" to treat the data in one extreme or the other (e.g., highly aged/spherical versus fresh/fractal). Reviewer #2 makes a number of very insightful comments regarding this aspect of the analysis, so I will not repeat many of her or his points here, however I would like to stress that this area of the analysis should be revisited, with particular attention paid to how uncertainties in the particle shape vs mixing affect the major conclusions of the paper.*

We have clarified the text so that we no longer assume any particle shape, but rely on the measured parameters (other replies below).

*There are many locations in the manuscript where the authors could be more specific without forcing the reader to consult tables and/or supplementary material. The methods section, not supplementary material, should give important details such as*

*the scan time being 30 minutes or that the sample was dried (to what RH?). Was an impactor used upstream of the DMA to remove larger particles that would affect the inversion? In addition, it would be stronger if more specific numbers were used rather than more general terms like more, less, or different.*

Those details (scan time, RH and impactor) are now given in the methods section. Terms more, less and different have been replaced by numbers when possible.

*Specific comments (page-line)*

*1-18: I recommend the authors use a different term here than elemental carbon, which to most readers will be interpreted as the operationally defined measurement, when I think they simply mean to say that the BC particles are initially externally mixed and "pure BC" for lack of a better term.*

Elemental changed to pure (black carbon).

*2-2: I suggest a slight re-phrasing, since being a CCN is not necessarily a requirement for cloud processing (e.g., collision processes)*

Cloud processing part is now removed.

*2-10: "from" to "for"?*

Done.

*2-29: Avoid use of "truly", because "size" is not a well-defined term (diameter, equivalent diameter, mass, surface area?). Better to say "mobility diameter" as that is the property relevant to the DMA used here.*

Done.

*3-26: Suggest very briefly stating what is involved in the inversion without going into details placed in supplementary information to give reader a sense of what is being done here. For example, "Number concentrations measured by the SP2 were converted to rBC number distributions by accounting for charged particle fractions and the DMA transfer function using standard DMPS inversion methods (see Supplementary Material)". Note the inversion does matter for reporting averages like number fraction or volume fraction if they include all particles and not just those identified as +1 charged.*

This section is now updated so that more details are given about the inversion method. Average number fraction and volume fraction meant bin averages (bin averages independent of the inversion), and not averages over mobility sizes (number-weighted average would depend on the inversion).

*3-29: The manuscript should also cite our original HTDMA-SP2 methods paper here (McMeeking et al., 2011). I also recommend the authors mention other work where the SP2 has been placed behind different types of classifiers, such as the APM and/or CPMA, such as Ohata et al. (2016).*

These papers are now cited.

*4-1/3: Another important thing to note here is higher output concentrations going through only one DMA instead of the two used in the VTDMA and HTDMA studies, allowing for improved counting statistics and faster scan times.*

The SP2 in the VTDMA study is actually measuring from the output of the first DMA, which means that they have equally good counting statistics with our method.

*4-23: Please clarify: I think the authors are referring to the SP2 size resolution? They should also mention a range of output sizes would be expected due to the DMA transfer function and that would have to be accounted for. It would also be useful to cite an*

*example from literature supporting the claim that most assume a narrow distribution of rBC core sizes...I'm not sure this is really the case.*

We have clarified the text so that the size resolution refers to the DMA. Width of the DMA transfer function is now given (FWHM about 45 nm for the 360 nm mobility size bin). The text about the assumed rBC core size distribution has been reformulated.

*Section 3.1 – it would be useful to have the average BC concentrations at each site given here, along with variability metric. Also better to state which of the sites was higher than saying there was a difference between them. Finally the measurements were performed at different times, so it would be helpful to know if there is any general trend in concentrations during the region over this time using the Aethalometer data.*

Average rBC concentrations and their variabilities are now given, and the location of the sites are now clarified. Unfortunately, we don't have Aethalometer data from Gual Pahari. Since our previous studies have shown that the sites have different annual cycles, Mukteshwar data should not be used alone. At least current measurements seem to be dominated by rapid variations rather than long term trends.

*5-7: I found this confusing. What is really meant by "homogenous" here...the authors mention multiple times in the paper that BC has different mixing states and morphologies, which could both affect the rBC vs DMA size relationship. Please elaborate.*

This bad example is now removed (better example given at the beginning of the results section) and we focus on describing Fig. 3.

*5-18: It might be possible to determine whether the larger particles are indeed multiply charged particles based on the rough sizing metrics available from the SP2 itself (and could be further checked against known charging fractions).*

Our aim was to say that larger particles have negligible contribution to the mobility size

bin mean values used in further calculations. For this it is irrelevant if these larger particles were multiply charged (they were) or pure compact rBC (such particles were not found as described in Sect 3.6).

*7-3: "it is expected that particles are spherical" Can the authors support this statement? I'm not sure if they meant "assumed" instead of "expected", since they caveat the entire statement with "For simplicity, ..." Is this section exploring a hypothetical exploration of the data, or real? Unless there is any strong evidence that particles are spherical at these sites, especially the rBC particles, than suggest re-phrasing to "For simplicity, we treat particles as spherical...". See general comments above.*

This section has been clarified. We are not expecting a particle shape, but report our directly measured parameters (e.g. rBC volume equivalent diameters and mobility diameters).

*7-13: Sulfate is not really considered semi-volatile under typically atmospheric conditions... Further, primary semi-volatile organics would, if anything, evaporate as concentrations decrease away from the urban areas leading to increases in the rBC volume fraction, though this could be counter-acted by formation of additional secondary semi-volatiles. I think "secondary" would be more accurate.*

Changed to "secondary".

*7-21: Are these numbers for total particles or just for the selected mobility diameter. Is the comparison to the literature values cited just before this statement describing the same (e.g., all or size-selected)?*

It is now clarified that this is for size-selected (360 nm) aerosol. Also, the different size ranges are now given.

*7-23/24: Is there any evidence that BB is the dominant source? Biomass burning (de-*

*pending on type, fuel, efficiency) will also emit non-rBC containing particles. Could other sources still be important? These might emit higher number fractions of rBC-containing particles compared to biomass burning. Were there any patterns in week-day versus weekend traffic or activities in these regions that might affect the sources?*

We don't have direct evidence of the dominant source, so now we now just mention regional BC emissions as an explanation for the high rBC particle number fraction. Our data set is too short for identifying weekday/weekend patterns (measurements took about six weeks in each location). Diurnal cycles are really the only patterns that we can see in our data.

*8-15/16: I agree with the vertical mixing affecting aerosol concentrations, but don't see how dilution would affect the rBC number fraction, which would be independent of concentration, unless some aerosol micro-physical processes are going on. Is there evidence that the aerosol outside the nocturnal boundary layer has lower rBC number fractions and mixes into the surface layer?*

rBC number fraction should decrease with altitude, because all major rBC sources are at the surface, but secondary organic aerosol formation can take place at any altitude.

*8-21: It is not clear whether the authors are comparing the two sites here, or the periods when the Gual Pahari or polluted Indo-Gangetic plains particles affect the site to those when there is little impact?*

We clarify that we are comparing the two sites (Gual Pahari and Mukteshwar).

*9-3: Please be more specific about what is meant here by the phrase "current experiments"...is this recent work in the field or specific to the Indian measurements?*

We clarify that these are the India measurements.

*9-9/11: I think this particular section represents the general problems with how the treatment of the uncertainties in the different size classification methods affects the conclusions of the paper. How was the cutoff of the 10% difference determined to know a particle was non-spherical? Does it account for an expected range of rBC and nonrBC refractive indices, which will also affect the difference between optical and mobility diameters? The authors should also provide the uncertainty in the LEO method for their instrument based on a comparison of the full Gaussian scattering measurements and LEO results for non-rBC containing results to support criteria further.*

The 10% limit was based on the SP2's sizing uncertainty, which is now clarified (also now giving a more conservative 20% limit). We have improved the description of the LEO results and included uncertainty analysis based on the range of refractive index values. LEO results are also compared with those without LEO for non-rBC particles.

*9-31/10-20: Unless I have missed something I believe the analysis described here treats the rBC particles as purely rBC particles to calculate the mass fractal dimensions? This contradicts earlier discussions in the paper regarding volume fraction. It would be better to restrict the calculations to particles likely to have less other material (e.g., smaller LEO size, maximum rBC mass observed in each DMA mobility size). I don't see an easy way to distinguish the shape effect on DMA sizing from the effects of other materials mixed with the rBC if this is not done.*

This part of the text has been removed.

*Figure 4: Remove the spurious low point that is related to the gain-stage matching in the SP2. Better to show as a gap than as a non-realistic lower data point.*

Done.

---

## Author Comment (AC4) · 25 Oct 2016

We would like to thank Cenlin He for the short comment. The comments below are shown with italicized font and our replies with the upright font. The changes to the manuscript and Supplementary Material are given as a supplement with replies to Referee #1.

*I have three short comments:*

*1. Page 1, Line 20: "..., but this also depends on the structure of the particle." Here, the authors missed several more recent studies on the morphological effects on BC optical properties. For example, He et al. (2015, 2016) showed that different structures for*

*both fresh and coated BC particles can lead to substantial variations in BC absorption and scattering. I suggest including these two studies as references here.*

*References:*

*He, C., Liou, K.-N., Takano, Y., Zhang, R., Levy Zamora, M., Yang, P., Li, Q., and Leung, L. R.: Variation of the radiative properties during black carbon aging: theoretical and experimental intercomparison, Atmos. Chem. Phys., 15, 11967-11980, doi:10.5194/acp-15-11967-2015, 2015.*

*He, C., Takano, Y., Liou, K.-N., Yang, P., Li, Q., and Mackowski, D. W.: Intercomparison of the GOS approach, superposition T-matrix method, and laboratory measurements for black carbon optical properties during aging, J. Quant. Spectrosc. Radiat. Transfer, in press, 2016.*

These two references have been added.

*2. Page 2, Line 17: "..., anthropogenic emissions such as biomass burning ..." Typically, biomass burning is referred to wildfire emission, which is not included in the anthropogenic (i.e. fossil fuel and biofuel burning) emissions. Here, are you referring to agricultural burning? Please clarify.*

Biomass is replaced by biofuel.

*3. Page 9, Lines 12–25: as stated in my first comment, recent studies (He et al., 2015, 2016) have shown that nonspherical/fractal structures of both fresh and coated BC particles can significantly affect BC optical properties and hence optical size during measurement. This could introduce large uncertainty into the optical method. The authors also showed results suggesting highly fractal BC aggregates in the measurement. In addition, the authors assumed the core-shell structure to quantify BC mixing state, which could also bring some uncertainty to their results due to the irregular coating structures in the real atmosphere. Thus, I suggest adding some discussions on how*

*fractal aggregating structures could possibly increase the uncertainty in measurements and analysis.*

We have added discussion about optical sizing uncertainty. Even with these uncertainties, there is a clear difference between optical and mobility sizes, which shows that rBC-containing particles are not spherical. Due to these sizing uncertainties (both optical and mobility), we are not anymore trying to estimate particle compositions.

---

## Author Response (AR2)

We would like to thank the Editor and the two Referees for the additional comments. Referee comments from the Editor report are shown below with italicized font and our replies including the updates to the manuscript are shown after each comment. A marked-up manuscript version is also showing the updates.

***Referee #1:***

*The paper has been largely improved but these points are still not resolved. It may be published after these are considered.*

*1. You have done the back trajectory analysis however have not found anything interesting, which means the sources are more local or basically very mixed? Could we get a general idea how much BC could be from biomass burning or traffic? I can marginally see the phase transition from the shift of BC features, is it at least associated with air mass coming from easterly or westerly?*

Our back trajectory analysis showed that most trajectories are originating from North West sector not far from the measurement sites. This means that the air masses are well mixed, but it cannot rule out local emissions. Since there is nothing new in this, we have left out the trajectory analysis. We were not able to separate particles from biomass burning and traffic. We are not sure what the Referee means by "the phase transition from the shift of BC features". For us the only statistically clear feature is the diurnal cycle.

*2. Is the very large BC maybe because of the dust? The GMD for Gual looks too large to me.*

We have examined single particle signals, but did not see any evidence of dust (e.g. clear differences in wide band to narrow band incandescence ratios). This is already mentioned in the manuscript. Gual Pahari GMD is larger due to the larger fraction of large rBC particles. Also note that volume mean diameter is larger than number mean diameter.

*3. It is better to use image plot for Fig. 7. and with colour scaled legend. It looks the bottom panel also has the second mode just been hidden by the other too populated mode.*

We ended up using a linear grayscale, because we wanted to show the relevant importance of the different modes, which is in line with reporting average values. A non-linear color scale could highlight the second mode in the bottom panel, but these are a small fraction of all particles and therefore are not that important for the averages. A legend has been added. For that we have normalized the data so that both cover the same range (from 0 to 1). Changes to the text and Fig. 7 are shown below.

Page 10, lines 4-6: "Figure 7 shows the dependency of the optical size on the rBC core size by means of a normalized probability density map (normalized by the maximum probability density)(darker color means higher probability)."

Figure 7, caption: "Distributions of single particle optical and rBC core diameters for the 360 nm mobility size and 30–35 °C instrument temperature range (probabilities normalized by the maximum values)(darker color indicates higher probability)."

Figure 7, original (left) and updated (right):

[Figure]

*4. Regarding the optical/mobility diameter of BC and BC core, could we have more details about Dm vs Do. The Dc/Dm tells the mixing state and Dm/Do tells the morphology.*

We are unsure of what kind of details would be needed, but there is not much more that could be provided. Time series, for example, are incomplete due to the problems with the LEO method at instrument temperatures below 30 °C. Also, we are not showing size-resolved LEO results, because optical sizes have a narrow sizing range (the detection limit is about 200 and the calibration extends up to about 450 nm) compared with that of the rBC cores.

*5. The bottom line of this paper is still not very conclusive, for example, you could have answered the questions: why distinct features of BC from two places, in terms of BC mixing state, size, and morphology. It's not really getting the punching point yet.*

We have used this data to show the distinct features of rBC from two places, but the data itself does not explain why they are distinct. In the absence of additional data that could link our observations to the sources and because trajectories could not provide this information, we have given up from the source analysis.

*Referee #2:*

*The authors have addressed most of the reviewer comments adequately, and the organization and presentation of the results is much clearer in the revised version. I would feel more confident in recommending publication if the authors were able to find the causes of several somewhat odd features of the data, such as the SP2 over counting and the temperature dependences. They are, however, quite transparent in the treatment of these issues, and the rarity of data in this region coupled with the interesting DMA -> SP2 are both strengths*

We have spent a lot of time in finding an explanation for the SP2 over counting and the temperature dependences. We have also made some experiments to quantify their effects (unpublished results), but we just don't have solid explanations for the causes. Nevertheless, the temperature dependency is a known issue and the over counting is a known issue for our SP2 that was used in the India measurements. Even if we could not confirm causes for these, we will clarify that these are known issues and there is a simple correction to the over counting and the weak temperature dependency is not a significant problem especially when we are focused on the rBC size.

[revised manuscript text omitted]